# Dynein-2 intermediate chains play crucial but distinct roles in primary cilia formation and function

Laura Vuolo[1], Nicola L Stevenson[1], Kate J Heesom[2], David J Stephens[1]*

[1]Cell Biology Laboratories, School of Biochemistry, University of Bristol, Bristol, United Kingdom; [2]Proteomics Facility, Faculty of Biomedical Sciences, University of Bristol, Bristol, United Kingdom

**Abstract** The dynein-2 microtubule motor is the retrograde motor for intraflagellar transport. Mutations in dynein-2 components cause skeletal ciliopathies, notably Jeune syndrome. Dynein-2 contains a heterodimer of two non-identical intermediate chains, WDR34 and WDR60. Here, we use knockout cell lines to demonstrate that each intermediate chain has a distinct role in cilium function. Using quantitative proteomics, we show that WDR34 KO cells can assemble a dynein-2 motor complex that binds IFT proteins yet fails to extend an axoneme, indicating complex function is stalled. In contrast, WDR60 KO cells do extend axonemes but show reduced assembly of dynein-2 and binding to IFT proteins. Both proteins are required to maintain a functional transition zone and for efficient bidirectional intraflagellar transport. Our results indicate that the subunit asymmetry within the dynein-2 complex is matched with a functional asymmetry between the dynein-2 intermediate chains. Furthermore, this work reveals that loss of function of dynein-2 leads to defects in transition zone architecture, as well as intraflagellar transport.
DOI: https://doi.org/10.7554/eLife.39655.001

*For correspondence:
david.stephens@bristol.ac.uk

Competing interests: The authors declare that no competing interests exist.

## Introduction

Cytoplasmic dyneins are minus-end directed motors that use the energy of ATP hydrolysis to move along microtubules. Two cytoplasmic dyneins have been identified. The better-characterized dynein-1 is involved in the transport of cargos in the cytoplasm, organelle dynamics and in mitotic spindle organization during mitosis (*Roberts et al., 2013*). Dynein-2 is responsible for retrograde transport in cilia and flagella. Primary (non-motile) cilia are hair-like extensions present on almost all animal cells that act as antennae for extracellular signals and are fundamental to proper metazoan development and ongoing health. They integrate signals in key pathways including sonic hedgehog (Shh), Wnt and platelet-derived growth factor signaling and participate in metabolic control and autophagy (*Reiter and Leroux, 2017*). Cilia are particularly important to ensure correct Shh signaling during embryonic development (*Goetz and Anderson, 2010*; *He et al., 2017*). Defects in cilia are linked to many human diseases, known collectively as ciliopathies, including developmental disorders, neurodegeneration and metabolic diseases (*Reiter and Leroux, 2017*; *Yee and Reiter, 2015*).

Ciliogenesis is initiated in non-dividing cells by the docking of pre-ciliary vesicles with the mother centriole. The pre-ciliary vesicles fuse and then surround the mother centriole concomitant with the assembly of a series of protein modules that form a diffusion barrier separating the distal end of the mother centriole from the rest of the cell cytoplasm (*Garcia-Gonzalo et al., 2011*). A microtubule bundle, the axoneme, then extends from the centriole to allow cargo transport by the process of intraflagellar transport (IFT) (*Yee and Reiter, 2015*). A 'transition zone' (TZ) that separates the mother centriole from the main length of the axoneme forms a diffusion barrier for both soluble and membrane proteins at the base of the cilium (*Garcia-Gonzalo et al., 2011*; *Garcia-Gonzalo and*

**eLife digest** Almost all cells in the human body are covered in tiny hair-like structures known as primary cilia. These structures act as antennae to receive signals from outside the cell that regulate how the body grows and develops. The cell has to deliver new proteins and other molecules to precise locations within its cilia to ensure that they work properly. Each cilium is separated from the rest of the cell by a selective barrier known as the transition zone, which controls the movement of molecules to and from the rest of the cell.

Dynein-2 is a motor protein that moves other proteins and cell materials within cilia. It includes two subunits known as WDR34 and WDR60. The genes that produce these subunits are mutated in Jeune and short rib polydactyly syndromes that primarily affect how the skeleton forms. However, little is known about the roles the individual subunits play within the motor protein.

Vuolo et al. used a gene editing technique called CRISPR-Cas9 to remove one or both of the genes encoding the dynein-2 subunits from human cells. The experiments show that the two subunits have very different roles in cilia. WDR34 is required for cells to build a cilium whereas WDR60 is not. Instead, WDR60 is needed to move proteins and other materials within an established cilium. Unexpectedly, the experiments suggest that dynein-2 is also required to maintain the transition zone.

This work provides the foundations for future studies on the role of dynein-2 in building and maintaining the structure of cilia. This could ultimately help to develop new treatments to reduce the symptoms of Jeune syndrome and other diseases caused by defects in cilia.

DOI: https://doi.org/10.7554/eLife.39655.002

*Reiter, 2017*). Once established, cilia are maintained and operated by the process of IFT (*Hou and Witman, 2015*). IFT-B complexes (comprising of a core subcomplex of nine subunits (IFT88, −81, −74, −70, –52, –46, −27, −25, and −22) with five additional, peripherally-associated subunits (IFT172, −80, –57, −54, and −20)) undergo kinesin-2-driven motility from base to tip where the complexes are then reorganized prior to retrograde transport of IFT-A (comprising six subunits (IFT144, -140, -139, -122, -121/WDR35, and -43)) driven by dynein-2 (*Hou and Witman, 2015*; *Jensen and Leroux, 2017*). This process ensures the correct localization of receptors and signaling molecules within cilia and directs transduction of signals from the cilium to the rest of the cell.

The genes encoding subunits of the dynein-1 and dynein-2 motors are largely distinct. Some light chain subunits are common to both motors but the major subunits (heavy, intermediate and light intermediate chains) are different between the two holoenzyme complexes. Dynein-2 is built around a heavy chain dimer of DHC2/DYNC2H1 (*Criswell et al., 1996*; *Mikami et al., 2002*). This associates with two intermediate chains (ICs), WDR34 and WDR60, first identified as dynein-2 subunits named FAP133 and FAP163 in *Chlamydomonas* (*Patel-King et al., 2013*; *Rompolas et al., 2007*) and subsequently shown to be components of metazoan dynein-2 (*Asante et al., 2013*; *Asante et al., 2014*). This asymmetry distinguishes dynein-2 from dynein-1 where two identical IC subunits form the holoenzyme. The reason for this asymmetry is unclear. In addition, a dynein-2-specific light intermediate chain (LIC3/DYNC2LI1) has been identified (*Hou and Witman, 2015*; *Mikami et al., 2002*) as well as a specific light chain, TCTEX1D2 (*Asante et al., 2014*; *Schmidts et al., 2015*).

Mutations in genes encoding dynein-2 subunits are associated with skeletal ciliopathies, notably short rib-polydactyly syndromes (SRPSs) and Jeune asphyxiating thoracic dystrophy (JATD, Jeune syndrome). These are recessively inherited developmental disorders characterized by short ribs, shortened tubular bones, polydactyly and multisystem organ defects (*Huber and Cormier-Daire, 2012*). In recent years, whole exome-sequencing technology has enabled the identification of new mutations involved in skeletal ciliopathies, notably a range of mutations affecting DYNC2H1 (DHC2, [*Chen et al., 2016*; *Cossu et al., 2016*; *Dagoneau et al., 2009*; *El Hokayem et al., 2012*; *Mei et al., 2015*; *Merrill et al., 2009*; *Okamoto et al., 2015*; *Schmidts et al., 2013a*]). Additionally, mutations in WDR34 (*Huber et al., 2013*; *Schmidts et al., 2013b*), WDR60 (*Cossu et al., 2016*; *McInerney-Leo et al., 2013*), LIC3/DYNC2LI1 (*Kessler et al., 2015*; *Taylor et al., 2015*) and TCTEX1D2 (*Schmidts et al., 2015*) have also been reported. The role of the dynein-2 heavy chain has been extensively studied in *Chlamydomonas*, *C. elegans*, and mice. In all cases, loss of dynein heavy chain

results in short, stumpy cilia that accumulate IFT particles at the tip, consistent with the role of dynein-2 in retrograde ciliary transport (*Hou and Witman, 2015*). Recently, more interest has been focused on the role of the subunits associated with DHC2/DYNC2H1. Two studies in *Chlamydomonas* and in human patient-derived fibroblasts revealed that LIC3/DYNC2LI1 (D1bLIC in *Chlamydomonas*) plays an important role in ciliogenesis and stabilization of the entire dynein-2 complex (*Li et al., 2015*; *Taylor et al., 2015*). Similarly, loss of Tctex2b (TCTEX1D2) destabilizes dynein-2 and reduces IFT in *Chlamydomonas* (*Schmidts et al., 2015*).

Previous work from our lab and others has shown that loss of function of dynein-2 intermediate chains, WDR34 and WDR60, is associated with defects in ciliogenesis. Knockdown of WDR60 or WDR34 in hTERT-RPE1 cells results in a reduction of ciliated cells, with an increase in length of the remaining cilia, likely depending on depletion efficiency (*Asante et al., 2014*). Mutations in WDR34 have also been shown to result in short cilia with a bulbous ciliary tip in patient fibroblast cells affected by SRP (*Huber et al., 2013*). Consistent with the results obtained in patient cells, loss of WDR34 in mice also results in short and stumpy cilia with an abnormal accumulation of ciliary proteins and defects in Shh signaling (*Wu et al., 2017*). Similarly, mutations in WDR60 patient fibroblasts are associated with a reduction in cilia number, although the percentage of ciliated cells was variable in different affected individuals (*McInerney-Leo et al., 2013*). These findings are all consistent with roles for WDR34 and WDR60 in IFT. Moreover, further recent data found that WDR60 plays a major role in retrograde ciliary protein trafficking (*Hamada et al., 2018*).

In this study, we sought to better understand the role of dynein-2 in human cells using engineered knockout (KO) cell lines for WDR34 and WDR60. We define a functional asymmetry within the complex, where WDR34 is absolutely required for cilia extension, while WDR60 is not. Loss of either IC results in defects in ciliary transition zone assembly and/or maintenance. We also find that loss of WDR60 leads to reduced assembly of the dynein-2 holocomplex and reduced interactions with IFT particles. Surprisingly, WDR34 is not required for the other dynein-2 subunits to assemble, instead loss of WDR34 results in delocalization of dynein-2 from the ciliary base and an accumulation of IFT proteins at this site. These data are consistent with a model in which WDR34 configures the dynein-2 complex for dynamic assembly and disassembly with IFT proteins to facilitate axoneme extension.

## Results

### WDR34 or WDR60 play different roles in cilia function

To understand the function of WDR34 and WDR60, we generated KO human telomerase-immortalized RPE1 (hTERT-RPE1) cells using CRISPR-Cas9. We derived two WDR34 KO clones (1 and 2) using guide RNAs (gRNAs) targeting exons 2 and 3, and one KO clone for WDR60, targeting exon 3. Genomic sequencing of these clones identified insertion/deletion mutations on the targeted sequences (*Figure 1—figure supplement 1*). All cell clones were analyzed for protein expression by immunoblot using polyclonal antibodies against multiple epitopes to exclude the possibility of downstream initiation sites being used. Neither WDR34 nor WDR60 was detected in the respective KO cells compared to the controls (*Figure 1—figure supplement 2*). To mitigate against the possibility of any off-target effects, we grew KO cells alongside control CRISPR cells which had been transfected with Cas9 and gRNA in the same way as the KO lines but showed no mutation at the target site. These cells (WDR34 KO CTRL and WDR60 KO CTRL) did not present any cilia defects when stained with Arl13b or IFT88 (*Figure 1—figure supplement 2*) and no differences were seen between these cells and wild type (WT) RPE-1 in any assay. Images in all figures show WT cells where indicated but indistinguishable results were also obtained using the control cell lines. Defects in ciliogenesis in both WDR34 and WDR60 KO cells were rescued by overexpressing WT proteins, confirming that the phenotypes we observed were not due to off-target mutations (described below).

Loss of WDR34 severely impaired the ability of the KO cells to extend a microtubule axoneme (*Figure 1A,B*), although Arl13b localized within those few cilia that did form. In contrast, loss of WDR60 did not significantly affect the ability of cells to extend an axoneme (*Figure 1B*) but did result in a change in localization of Arl13b to a more pronounced enrichment at the base and tip of many cilia compared to the more uniform ciliary distribution in control cells. Cilia were shorter in both WDR60 and WDR34 KO cells (*Figure 1C*). Next, we examined the assembly and structure of

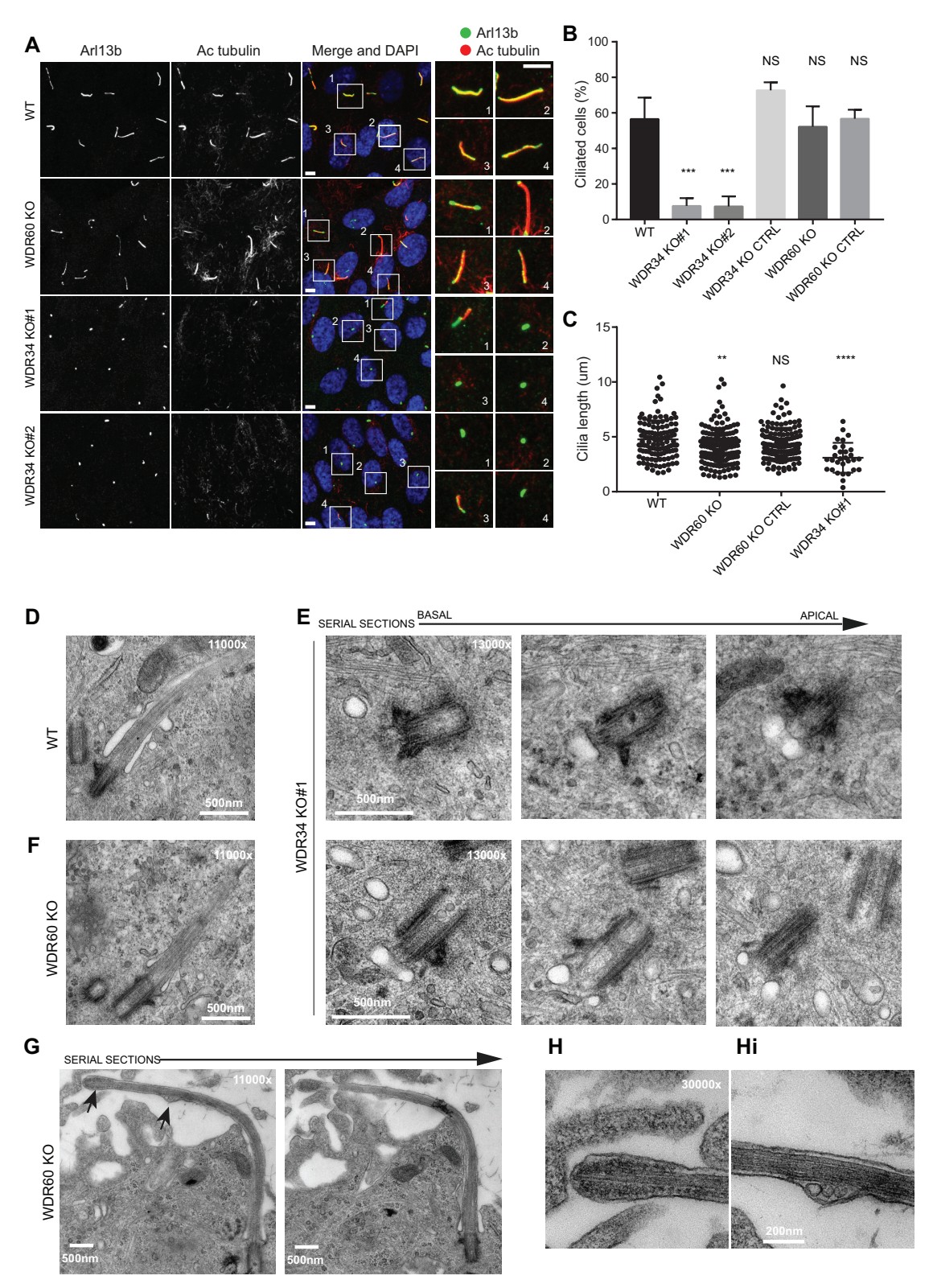

**Figure 1.** Role of dynein-2 intermediate chains WDR34 and WDR60 in ciliogenesis. (**A**) Cilia stained with the markers Arl13b (green) and acetylated tubulin (AcTub, red) in RPE1 WDR60 and WDR34 KO cell lines. Panels on the right show enlargements of the boxed regions. Scale bars 5 µm. (**B**) Percentage of ciliated cells (n = 3; 656 WT, 430 WDR34 KO#1, 296 WDR34 KO#2, 397 WDR34 KO CTRL, 343 WDR60 KO and 195 WDR60 KO CTRL cells quantified). (**C**) Cilium length in WDR60 and WDR34 KO compared with WT cells and CRISPR control cells lines (n = 3; 120 WT, 158 WDR60 KO, 138

*Figure 1 continued on next page*

*Figure 1 continued*

WDR60 KO CTRL and 30 WDR34#1 cells quantified). Mann-Whitney test was used, p-value: ****=<0.0001. (D–Hi) Representative 70 nm thick EM sections of (D) WT, (E) WDR34 KO and (F, G– Hi) WDR60 KO cilia. (E) Six serial sections through a WDR34 KO cilium showing no axoneme extension. (G) Two serial sections through a WDR60 KO cilium showing complete cilium. Arrows point to the bulbous ciliary tip and to a membrane protrusion containing membrane vesicles; enlargements are shown to the right (H and Hi). Scale bar length and magnification is indicated on the images.
DOI: https://doi.org/10.7554/eLife.39655.003

The following figure supplements are available for figure 1:

**Figure supplement 1.** Generation of a WDR60 and WDR34 KO cell line.
DOI: https://doi.org/10.7554/eLife.39655.004

**Figure supplement 2.** CRISPR control cells lines show no defect in ciliogenesis.
DOI: https://doi.org/10.7554/eLife.39655.005

primary cilia in WDR34 and WDR60 KO cells by transmission electron microscopy (EM). After 24 hr of serum starvation, WT RPE1 cells extend a defined axoneme surrounded by a ciliary membrane (*Figure 1D*). In contrast, WDR34 KO cells failed to extend an axoneme (*Figure 1E*) but showed a large docked pre-ciliary vesicle, consistent with the small Arl13b-positive structures seen by immunofluorescence. WDR60 KO cells showed apparently normal cilia (*Figure 1F*) with normal basal body structures and axoneme extension. However, when an entire cilium was captured in WDR60 KO serial sections (*Figure 1G*), we observed a bulged cilium tip containing accumulated electron dense particles (*Figure 1H*). To our surprise, we also observed the ciliary membrane bulged at a second point along the axoneme and this region contained intraciliary vesicular structures (*Figure 1Hi*).

## Loss of WDR34 and WDR60 causes accumulation of proteins at the ciliary tip

The abnormal structure of cilia in the KO cells led us to analyze the steady-state localization of the IFT machinery. After 24 hr serum starvation, IFT88 (part of IFT-B) was found almost exclusively at the base of the cilia in WT RPE1 cells but in WDR60 KO and WDR34 KO cells IFT88 was found throughout the cilia and accumulated at the tips (*Figure 2A and B*, quantified in *Figure 1Ai*). A similar phenotype was observed for IFT-B components, IFT54 (*Figure 2C*) and IFT57 (*Figure 2D*, quantified in *Figure 2Di*). In those WDR34 KO cells that did extend cilia, IFT-B proteins were seen to accumulate at the tip of cilia. The limited number of ciliated WDR34 KO cells precluded further quantification. Another IFT-B protein, IFT20 (*Figure 2E*), was enriched at both the tip and the base of cilia in WDR60 KO cells (quantification in *Figure 2F*). IFT20 is the only IFT component found to localize to the Golgi until now. As expected, in WT cells IFT20-GFP was found at the ciliary base with a ribbon-like distribution, consistent with its known localization to the Golgi. This pool of IFT20-GFP was largely absent from WDR60 KO cells (*Figure 2G*). We have tried to measure IFT in these cells using a variety of markers including Arl13b- and IFT88-fusions but have not been able to derive reliable quantitative data.

Next, we analyzed the localization of IFT-A proteins. We found that both IFT140 (*Figure 3A*) and IFT43 (*Figure 3B*) were accumulated along the length of the axoneme within cilia in WDR60 KO cells, as well as at the tips in the few cilia present in WDR34 KO cells, while they were found only at the base of the cilia in WT cells (quantification in *Figure 3Ai–iii* and 3Bi-iii). Further quantification showed that both IFT140 (*Figure 3C*) and IFT43 (*Figure 3D*) are enriched in WDR60 KO cilia compared to controls. In addition, we determined the localization of a subunit of the anterograde kinesin-2 motor, KAP3 which was also accumulated at the ciliary tip in WDR60 KO cells (*Figure 3E*).

To study whether defects in dynein-2 affect the transport of membrane proteins we used GFP-fusions with Arl13b, somatostatin receptor type 3 (SSTR3), 5-hydroxytryptamine receptor type 6 (5HT6) and Rab8a. We found that in live cells Arl13b-GFP and EGFP-SSTR3 localize at the ciliary tip in WDR60 KO cells, which appear enlarged and bulbous in these cells, consistent with EM data (*Figure 4A and B*). We also noticed a consistent reduction in the amount of Arl13b-GFP within cilia in WDR60 KO cells compared to WT cells (*Figure 4C and Ci*). The same observation was made with EGFP-SSTR3 (*Figure 4D and Di*) and EGFP-5HT6 (*Figure 4E and Ei*). In contrast, GFP-Rab8a localization in the cilia was indistinguishable between WT and WDR60 KO cells (*Figure 4F and Fi*).

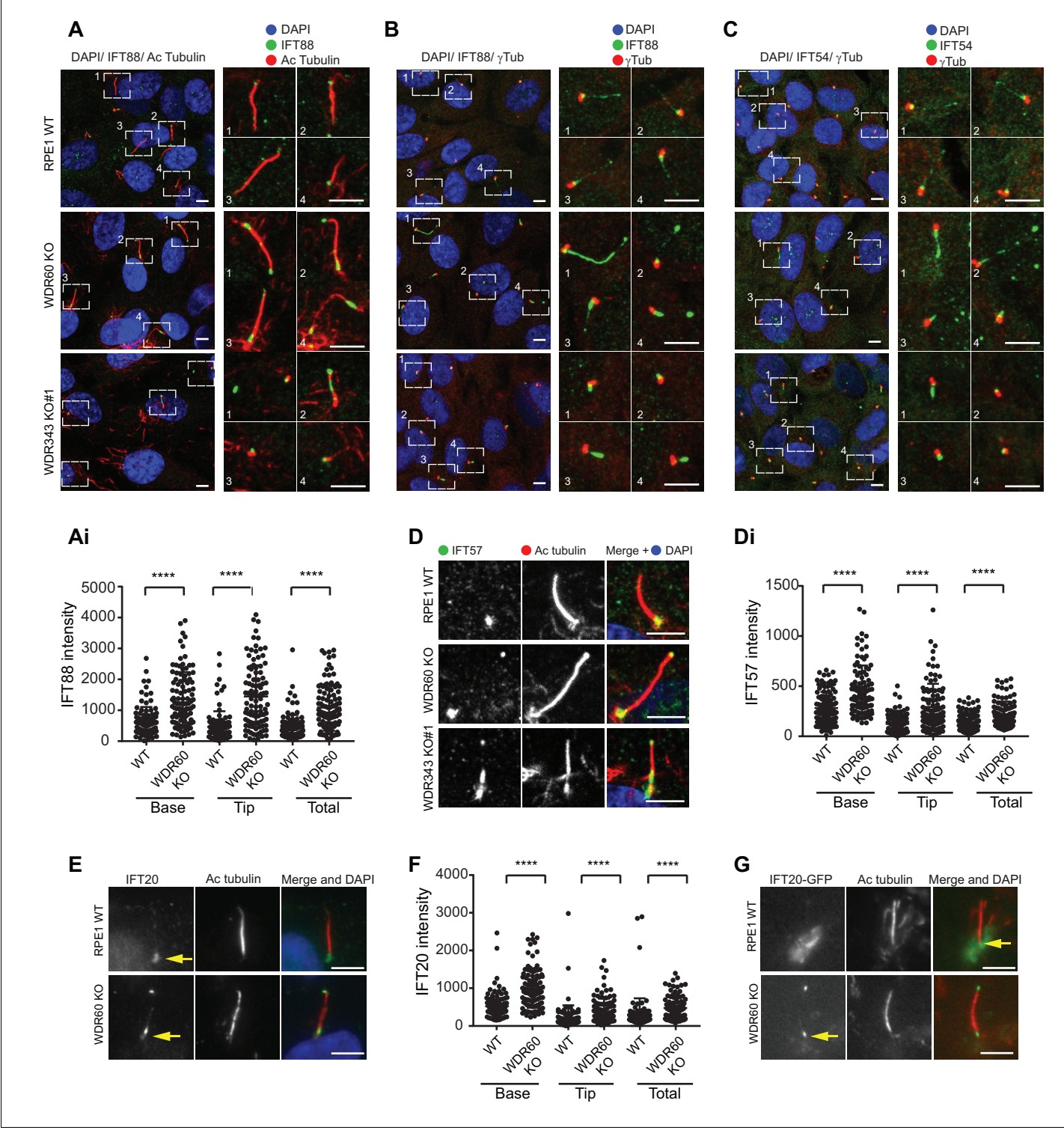

**Figure 2.** WDR34 and WDR60 are essential for IFT-B trafficking in primary cilia. (**A**) Localization of IFT88 in WT, WDR60 KO, and WDR34 KO#1 cells stained with the ciliary marker Acetylated tubulin. (**Ai**) Quantification of IFT88 localization within cilia in WT and WDR60 KO cells (102 WT and 101 WDR60 cells quantified). (**B and C**) Localization of IFT88 (**B**) and IFT54 (**C**) in WT, WDR60 KO, and WDR34 KO#1 cells stained with the centrosome marker gamma-tubulin. (**D**) Localization of IFT57 in WT, WDR60 KO, and WDR34 KO#1 cells stained with acetylated tubulin. (**Di**) Quantification of IFT57 within cilia in WDR60 KO and WT cells (IFT57 106 WT and 98 WDR60 KO cells quantified; n = 3 independent experiments). Mann-Whitney test was used, p-value: ****=<0.0001. (**E**) Endogenous IFT20 accumulates at the ciliary tip in WDR60 KO cells. (**F**) Quantification of IFT20 localization within cilia

*Figure 2 continued on next page*

*Figure 2 continued*

in WDR60 KO cells (n = 3 independent experiments, 102 WT and 100 WDR60 KO cells quantified). (G) Localization of IFT20-GFP in WT and WDR60 KO fixed cells. Scale bar, all panels = 5 μm. Arrows point to the ciliary base.

DOI: https://doi.org/10.7554/eLife.39655.006

## Dynein-2 is required for ciliary transition zone assembly

Previous studies have shown that the protein content of cilia is maintained by a diffusion barrier formed by the transition zone. Changes in transition zone composition have been associated with the mislocalization of ciliary proteins, including the membrane marker Arl13b (*Li et al., 2015*; *Shi et al., 2017*). To test if the reduction in Arl13b seen in our WDR60 KO was caused by a defect in the transition zone, we labeled KO and WT cilia with known transition zone markers. We found that the core transition zone marker, RPGRIP1L (also known as MKS5), is no longer restricted to an area adjacent to the mother centriole in WDR60 KO cells (*Figure 5A*, quantified in 5Ai). Conversely, TMEM67 (also known as MKS3), which in WT cilia extends from the basal body through a more distal region, becomes much more tightly restricted to the base of the cilium in WDR60 KO cells (*Figure 5B*, quantified in 5Bi). We also determined the transition zone organization in WDR34 KO cells. The few cilia found in the WDR34 KO cells recapitulate the same phenotype observed in the WDR60 KO cilia with an expansion of RPGRIP1L to a more distal position and a reduction of the TMEM67 domain (*Figure 5A* and *Figure 5B*). In contrast to TMEM67 and RPGRIP1L, no changes were observed for the transition zone marker TCTN1 in both WDR34 KO and WDR60 KO cells with respect to the control (*Figure 5C*).

## Unregulated entry of smoothened into cilia following loss of WDR60

Defects in the dynein-2 motor have been previously associated with deregulation of the Shh pathway (*May et al., 2005*). Smoothened (Smo), a key component of Shh signaling, localizes to the cilia in response to Shh stimulation but is normally excluded from cilia in cells that have not been treated with Shh or an equivalent agonist such as Smoothened agonist (SAG). Unexpectedly, we found that Smo was localized to cilia in WDR60 KO cells even in the absence of SAG stimulation (*Figure 6A and Ai*). This localization did not increase upon agonist treatment. In contrast, Smo was indeed excluded from cilia in WT cells at steady state (*Figure 6A and Ai*) but was readily detected within cilia following SAG stimulation (*Figure 6B, Bi, and Bii*).

## Expression of wild type and patient mutants of WDR60 and WDR34 in KO cells

Many mutations in WDR34 and WDR60 have been associated with SRPs and JATD syndromes. We engineered selected patient mutations into a WDR60 construct and expressed the proteins in WDR60 KO cells to see how well the mutated proteins rescue KO phenotypes compared to expression of WT proteins. Two WDR60 mutations were selected from one SRPS patient with compound heterozygosity for WDR60 (WDR60[T749M], WDR60[Q631*]) (*McInerney-Leo et al., 2013*). The first mutation is located in the WD repeat region (WDR60[T749M]) and the second (WDR60[Q631*]) is located just before the WD repeat domain (*Figure 7A*).

Protein expression of stably transfected WT and HA-WDR60 mutants in WDR60 KO cells is shown in *Figure 7—figure supplement 1*. Both WT HA-WDR60 and HA-WDR60[T749M] efficiently rescued defects in the localization of IFT88 (*Figure 7B*). However, expression of HA-WDR60[Q631*] was unable to restore the basal body localization of IFT88 (*Figure 7B and Bi*). Similar data were obtained for the localization of IFT140 (*Figure 7C and Ci*); expression of both WT and the HA-WDR60[T749M] mutant of WDR60 restored the localization of IFT140 to the base of the cilia, but IFT140 persisted throughout the cilium following expression of HA-WDR60[Q631*] as in WDR60 KO cells. Next, we tried to mimic the compound heterozygosity described in patient cells by generating a stable cell line expressing both WDR60 [T749M] and [Q631*] mutants (*Figure 7—figure supplement 1B*). When the two mutants were co-expressed in the same WDR60 KO cells we saw no additive or dominant negative effects, but cilia appeared normal with IFT88 only localized to the base (*Figure 7—figure supplement 1*), as was seen with the HA-WDR60[T749M] mutant rescue.

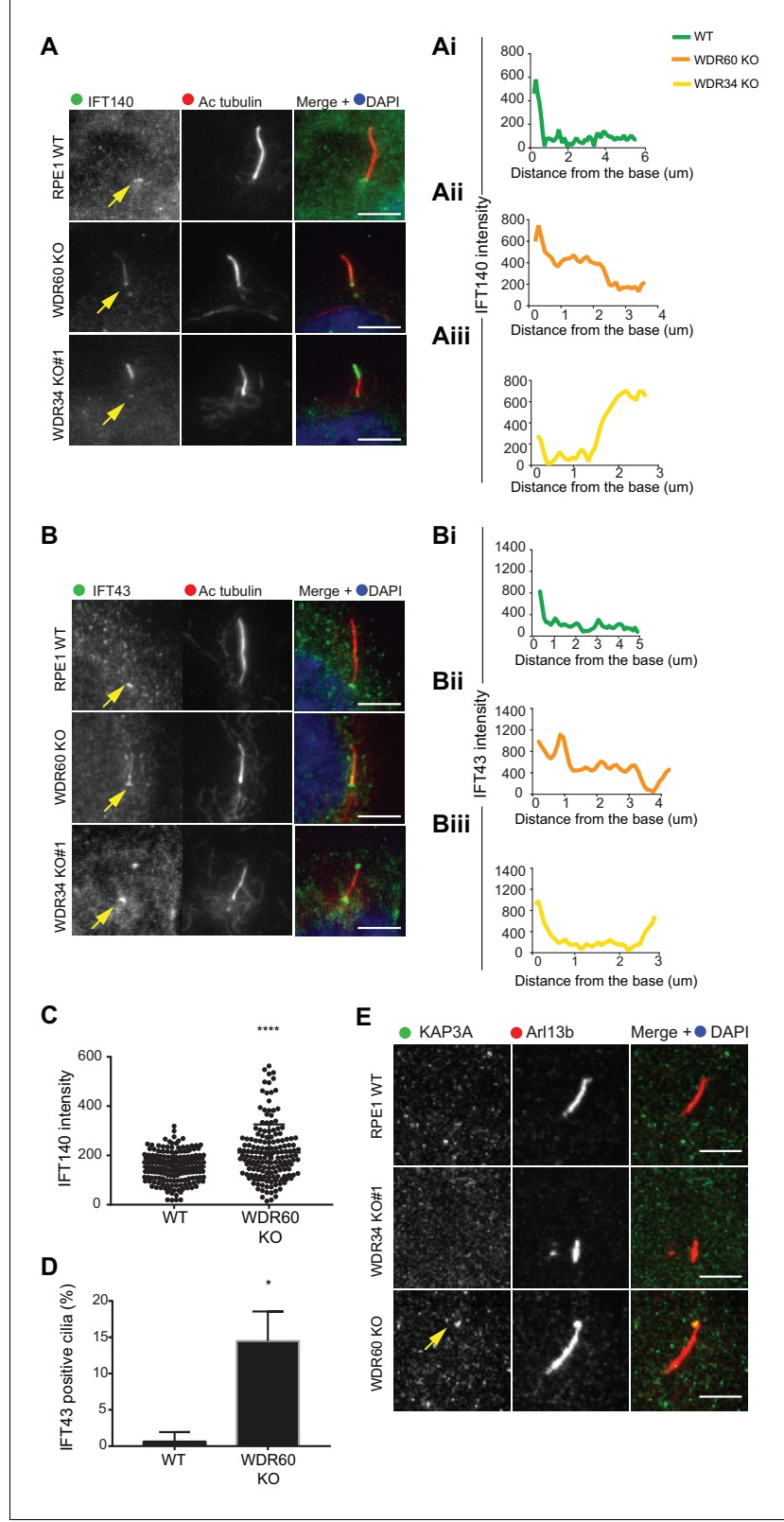

**Figure 3.** IFT-A trafficking defects in absence of WDR34 and WDR60. (**A–B**) Localization of IFT-A proteins (**A**) IFT140 and (**B**) IFT43 in WT, WDR60 KO, and WDR34 KO#1 cells. Line graphs show lines scans of IFT intensity along the length of a representative cilium from WT (Ai and Bi, green), WDR60 KO (Aii and Bii, orange) and WDR34 KO (Aiii and Biii, yellow) cells. (**C**) Quantification of IFT140 intensity within cilia in WT and WDR60 KO cells

*Figure 3 continued on next page*

*Figure 3 continued*

(n = 3, 186 WT and 166 WDR60 KO cells quantified). (D) Quantification of the number of IFT43 positive cilia from WT and WDR60 KO cells (n = 3, 271 WT and 203 WDR60 KO cells quantified). (C–D) Mann-Whitney test was used, p-value: ****=<0.0001. (E) Localization of KAP3A in WT, WDR34 KO, and WDR60 KO cells. Scale bars = 5 μm. Arrows point to the ciliary base.

DOI: https://doi.org/10.7554/eLife.39655.007

In parallel, we found that expression of WT mGFP-WDR34 restored ciliogenesis and axoneme extension in WDR34 KO cells (*Figure 7D*, quantified in *Figure 7Di*). Moreover, WT WDR34 was able to rescue IFT88 localization to the basal body (*Figure 7E*).

To better understand the function of WDR60 we analyzed how the dynein-2 complex is assembled in the presence of WT and mutant WDR60 proteins by performing immunoprecipitation. Immunoblotting using an anti-HA antibody showed that WT HA-WDR60 and HA-WDR60[T749M] (125 kDa) were expressed at similar levels to the truncated HA-WDR60[Q631*] mutant (75 kDa) (*Figure 7F*). We found that immunoprecipitation of HA-WDR60 expressed in WDR60 KO cells effectively pulls down the chaperone NudCD3, known to interact with dynein-2 via its WD repeat domains (*Asante et al., 2014*). As expected, NudCD3 did not bind to HA-WDR60[Q631*] lacking the WD repeat domain (*Figure 7F*). WT WDR60 also bound effectively to WDR34 and, notably, this interaction was very similar with the HA-WDR60[T749M] or [Q631*] mutants (*Figure 7G*). In contrast, LIC3/DYNC2LI1 was readily detected with WT WDR60 but less so with WDR60[T749M] and only a very small amount of LIC3/DYNC2LI1 was detected bound to WDR60[Q631*] (*Figure 7G*). TCTEX1/DYNLT1 was found to bind effectively to both WT and HA-WDR60[T749M] but less well to HA-WDR60[Q631*] (*Figure 7G*). Next, we tested the interactions with IFT proteins, the primary cargo of the ciliary motors. We found that WT WDR60 can bind to IFT140, IFT88, and IFT57; WDR60[T749M] binds to all 3 IFT subunits tested but binds less well to IFT140 (*Figure 7H*). In contrast, WDR60[Q631*] pulled down reduced levels of IFT88 and IFT57 and did not interact with IFT140.

## The stability of dynein-2 complex in WDR34 and WDR60 KO cells

It has been reported that loss of some components of dynein-2 modifies the stability of the whole dynein-2 complex. In *Chlamydomonas* depletion or loss of LIC3/DYNC2LI1 (D1bIC2) causes a reduction of DHC2/DYNC2H1 in whole cell lysate, whereas the expression level of the intermediate chain is less affected (*Reck et al., 2016*). Similar results were obtained analyzing expression levels of DHC2/DYNC2H1 in patients cells with LIC3/DYNC2LI1 mutations (*Taylor et al., 2015*). We have shown previously that siRNA depletion of WDR34 affects the stability of WDR60 and vice versa (*Asante et al., 2014*). To determine whether loss of one intermediate chain had an effect on the stability of the other, we analyzed levels of WDR34 and WDR60 in whole cells lysate of serum-starved KO cells. Notably, we found that in serum-starved cells, expression levels of WDR34 were reduced in WDR60 KO cells, although not completely lost. Correspondingly, there was a reduction of WDR60 expression levels in WDR34 KO whole cell lysate (*Figure 8A*).

Next, we sought to determine the effect of WDR60 and WDR34 loss on the localization of other dynein-2 subunits. We found that LIC3/DYNC2LI1 localized in cilia of WT cells, but this localization was lost in WDR34 or WDR60 cells (*Figure 8B*). DHC2/DYNC2H1 was detected at the base of the cilia in WT and WDR60 KO cells, but not along the ciliary axoneme. Interestingly, DHC2/DYNC2H1 localization at the ciliary base was reduced in WDR34 KO cells (*Figure 8C* and 8 Ci). TCTEX1 was enriched at the base of the cilium in all cell lines (DYNLT1, *Figure 8D*, and 8Di). To test how the loss of one dynein-2 intermediate chain affected the localization of the other we overexpressed HA-WDR34 and HA-WDR60 in WDR60 and WDR34 KO cells. Both HA-WDR34 and HA-WDR60 were enriched at the base and in the ciliary axoneme in WT cells (*Figure 8—figure supplement 1A*). We observed no changes in the localization of HA-WDR34 in WDR60 KO cells (*Figure 8—figure supplement 1B*) however, surprisingly, HA-WDR60 was greatly enriched in the stumpy cilia of WDR34 KO cells compared to the cilia in WT cells (*Figure 8—figure supplement 1A1*). Overexpression of HA-WDR60 could not rescue axoneme elongation in WDR34 KO cells (*Figure 8—figure supplement 1C*). Additionally, overexpression of HA-WDR34 could not rescue abnormal IFT88 localization in WDR60 KO cells (*Figure 8—figure supplement 1D*).

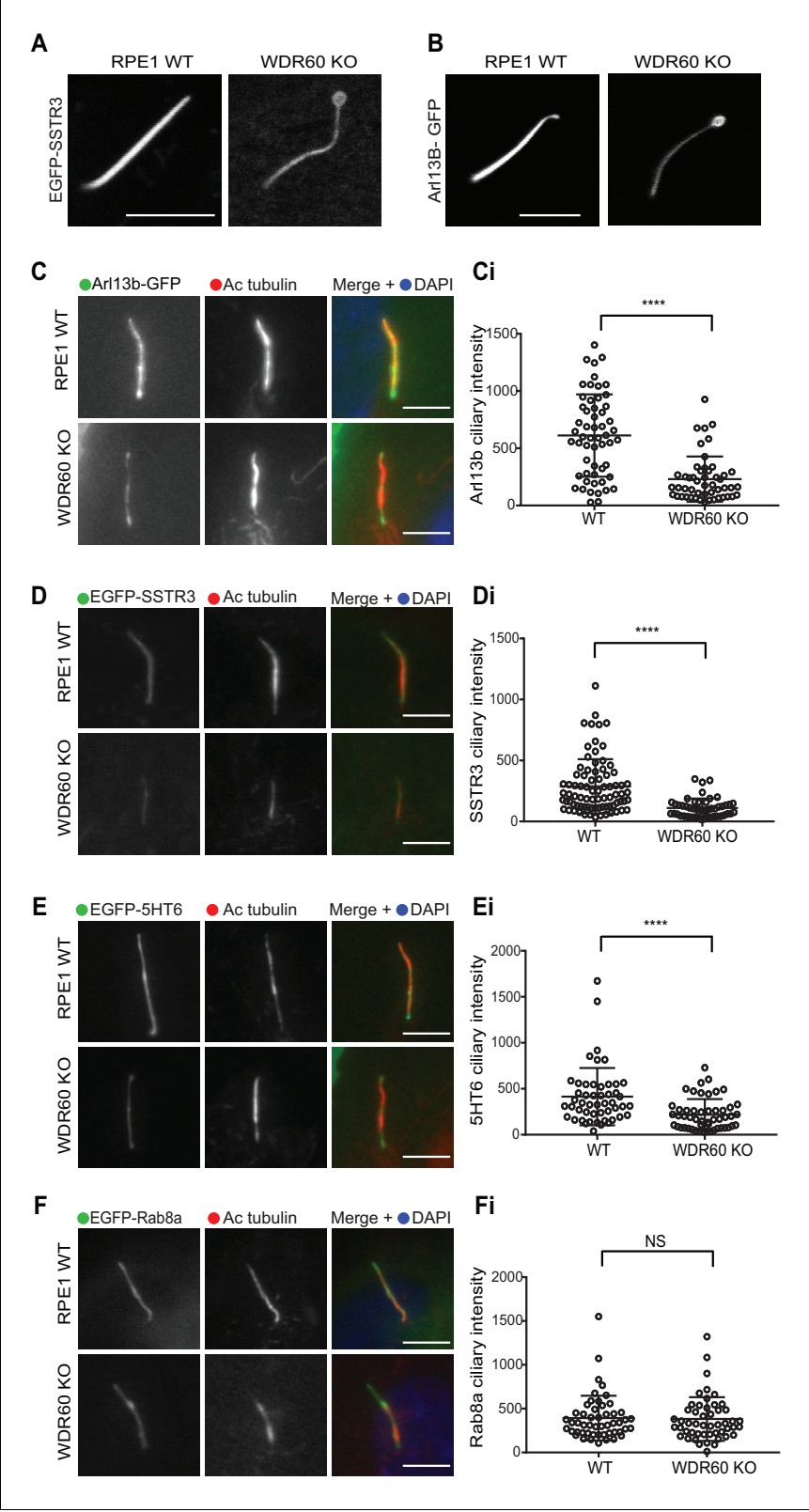

**Figure 4.** WDR60 is crucial for the composition of ciliary membrane proteins. (**A and B**) Single frame images taken from live imaging movies of WT and WDR60 KO cells overexpressing (**A**) EGFP-SSTR3 and (**B**) GFP-Arl13b. (**C–F**) Fixed cell staining of overexpressed (**C**) Arl13b-GFP, (**D**) EGFP-SSTR3, (**E**) EGFP-5HT6, and (**F**) EGFP-Rab8a in WT and WDR60 KO cells. (Ci-Fi) Intensity quantification of the overexpressed protein indicated (Arl13b-GFP n = 3, 56

*Figure 4 continued on next page*

*Figure 4 continued*

WT, and 50 WDR60 KO cells quantified; EGFP-SSTR3 n = 3, 80 WT and 50 WDR60 KO cells quantified; EGFP-5HT6 n = 3, 50 WT, and 51 WDR60 KO cells quantified; EGFP-Rab8a n = 3, 51 WT, and WDR60 50 KO cells quantified). Mann-Whitney test was used, p-value: ****=<0.0001. Scale bars 5 μm.
DOI: https://doi.org/10.7554/eLife.39655.008

## Loss of WDR34 stalls dynein-2 complexes coassembled with IFT proteins

The results described above suggest a defect in the assembly of the dynein-2 holoenzyme and therefore we used a proteomic approach to define the assembly of dynein-2. We stably expressed HA-WDR34 in WT and WDR60 KO cells and HA-WDR60 in WT and WDR34 KO cells and performed immunoprecipitations using HA-GFP as a control. Confirming that both WDR34 and WDR60 exist in the same complex, immunoprecipitation analysis showed that in WT cells HA-WDR60 pulls down WDR34, while HA-WDR34 pulls down WDR60 (*Figure 9—figure supplement 1*).

Multiplex tandem-mass-tag (TMT) labeling enabled us to define the interactome of WDR34 in the presence and absence of WDR60 and of WDR60 in the presence and absence of WDR34 (*Figure 9A* and Supplemental data file S6). No obvious candidate proteins emerged from this analysis that could explain the axoneme extension defect in WDR34 KO cells. Therefore we focussed on changes in interactions with known dynein-2 and IFT proteins. We found that the interactions of HA-WDR34 with dynein-2 components are reduced but not lost in WDR60 KO cells compared to WT cells (up to

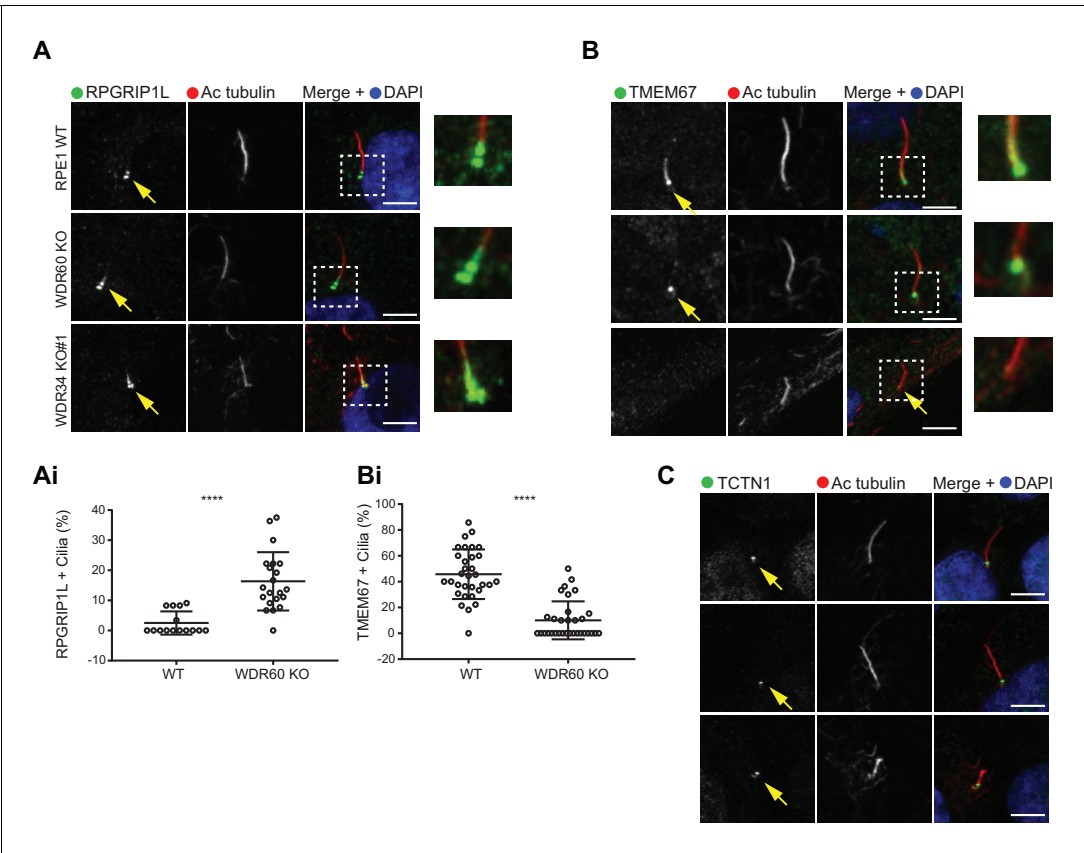

**Figure 5.** Dynein-2 is important for transition zone assembly. (A–C) Localization of (A) RPGRIP1L, (B) TMEM67, and (C) TCTN1 in WT and KO cells. Enlargements from the box regions are shown on the right. (Ai and Bi) Percentage of RPGRIP1L and TMEM67 positive cilia (RPGRIP1L n = 3, 188 WT, and 272 WDR60 KO cells quantified; TMEM67 n = 3, 359 WT, and 243 WDR60 KO cells quantified). Mann-Whitney test was used p-value: ****=<0.0001. Scale bars 5 μm. Arrows point to the ciliary base.
DOI: https://doi.org/10.7554/eLife.39655.009

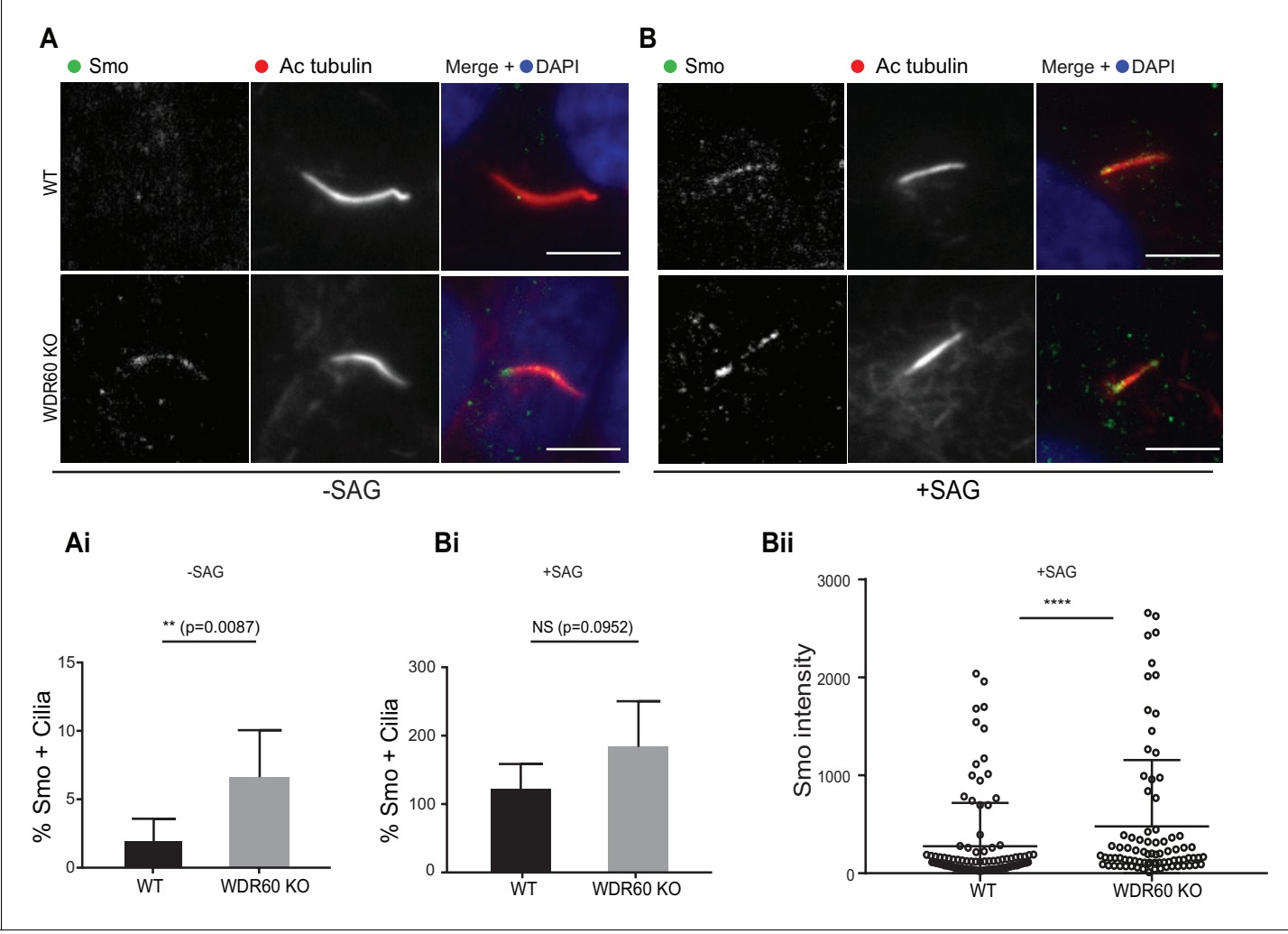

**Figure 6.** Loss of WDR60 affects Smo localization in the cilia. (**A and B**) Immunofluorescence of WT and WDR60 KO cells in presence or absence of SAG and stained for Smo (green), AcTub (red), and DAPI (blue). (**Ai and Bi**) Percentage of Smo positive cilia in SAG untreated (n = 3, 148 WT and 120 WDR60 KO cells quantified) and treated cells (n = 3, 670 WT and 580 WDR60 KO cells quantified). (**Bii**) Quantification of the total intensity of ciliary Smo labeling in cells treated with SAG (n = 3, 102 WT and 82 WDR60 KO cells quantified). Mann-Whitney test was used, p-value: ****=<0.0001. Scale bars 5 μm.

DOI: https://doi.org/10.7554/eLife.39655.010

three $\log_2$-fold change, *Figure 9B*). Notably, loss of WDR60 caused a reduction in the amount of DHC2/DYNC2H1 and TCTEX1/DYNLT1 associated with HA-WDR34 (*Figure 9B and C*). Moreover, interactions between HA-WDR34 and LIC3/DYNC2LI1, TCTEX3/DYNLT3, and the molecular chaperone NudCD3 were also reduced in WDR60 KO cells. Furthermore, we found a reduction in the amount of several IFT-B components (IFT57, IFT74, and IFT88) associated with HA-WDR34. Two IFT-A components (WDR19/IFT144 and WDR35/IFT121) and two BBSome components (BBS7 and BBS9) were identified to interact with HA-WDR34 and their binding to HA-WDR34 was reduced in the absence of WDR60. Among these components, the binding of WDR19/IFT144 with HA-WDR34 was the most affected by the loss of WDR60. This suggests a more loosely assembled dynein-2 motor that also shows reduced, but importantly still readily detectable, binding to IFT and BBSome proteins.

In contrast, and to our surprise, in the absence of WDR34, interactions of HA-WDR60 with other dynein-2 components, including DHC2/DYNC2H1, were only slightly reduced (<1 $\log_2$-fold change, *Figure 9B and D*). LIC3/DYNC2LI1, TCTEX1/DYNLT1, TCTEX3/TCTEX1L/DYNLT3, LC8-type 1/ DYNLL1, LC8-type 2/DYNLL2 and NudCD3 bound with similar efficiency to HA-WDR60 in WT and

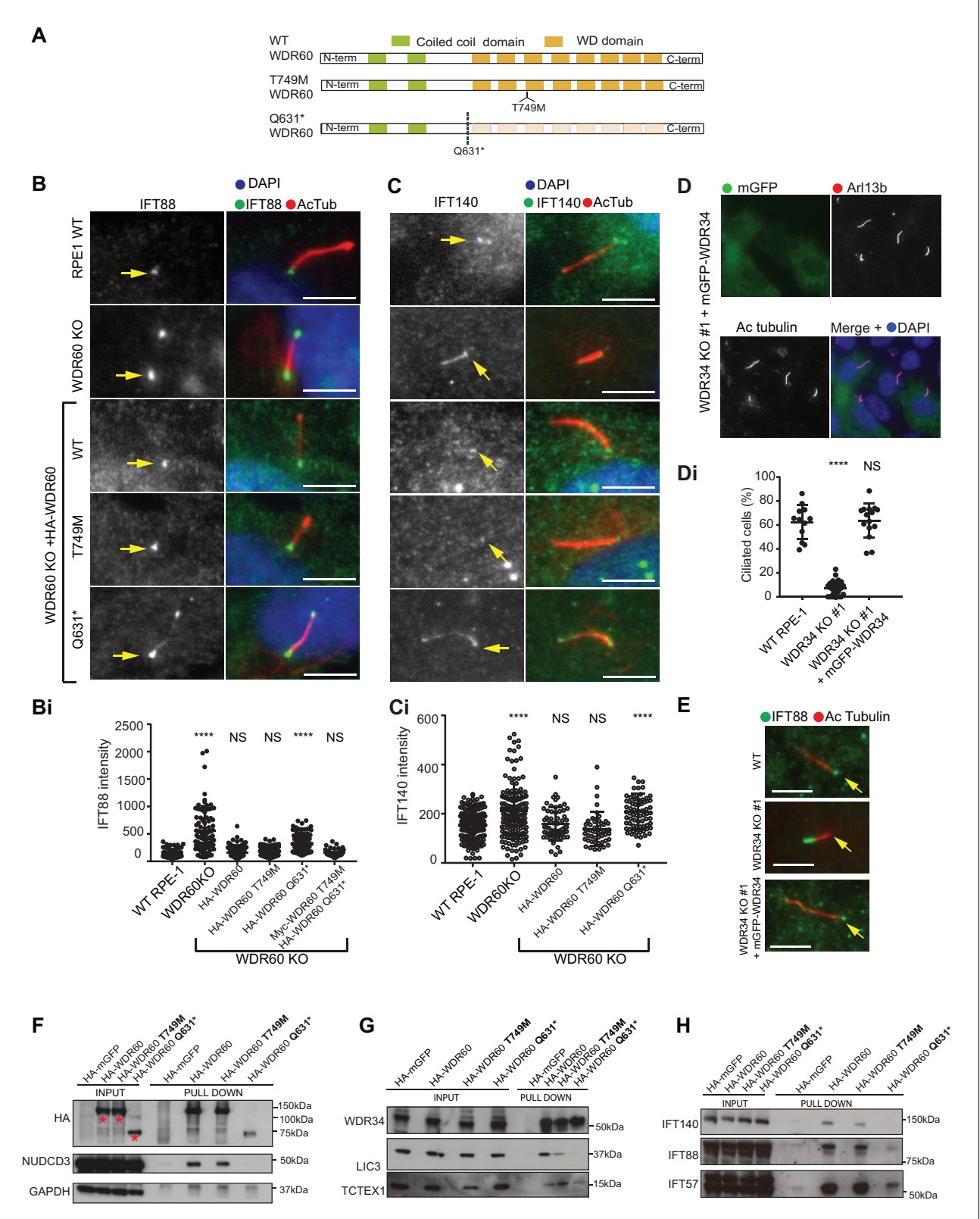

**Figure 7.** WDR34 and WDR60 KO rescue experiments. (**A**) Diagrams of WT and WDR60 mutant proteins structures. (**B and C**) IFT88 or IFT140 staining (green) with AcTub labeling (red) of the stable cell lines shown in *Figure 7—figure supplement 1A*, as well as WT cells. (**Bi**) Total intensity quantification of IFT88 labeling across the length of primary cilia in each cell line (n = 3; 97 WT, 125 WDR60 KO, 202 WDR60 KO+ HA-WDR60, 119 WDR60 KO+ HA-WDR60 [T749M], 150 WDR60 KO+ HA-WDR60 [Q631*] cells quantified). (**Ci**) Total intensity quantification of IFT140 labeling across the

*Figure 7 continued on next page*

*Figure 7 continued*

length of primary cilia in each cell line (n = 3; 168 WT, 164 WDR60 KO, 63 WDR60 KO+ HA-WDR60, 56 WDR60 KO+ HA-WDR60 [T749M], 71 WDR60 KO+ HA-WDR60 [Q631*] cells quantified). (D) WDR34 KO#1 cells stably expressing mGFP-tagged WT and mutant WDR34. Primary cilia staining with Arl13b (red) and AcTub (blue) (Di) Percentage of ciliated cells in WT, WDR34 KO#1 cells and WDR34 KO#1 cells stably expressing mGFP-WDR34 (n = 3; 357 WT, 430 WDR34 KO, 399 WDR34 KO+ mGFP-WDR34 cells quantified). (E) IFT88 staining in WT, WDR34 KO#1 cells, and WDR34 KO#1 cells expressing GFP-tagged WDR34. One-way ANOVA followed by Kruskal-Wallis test was used p-value: ****=<0.0001. Scale bars 5 μm. Arrows point to the ciliary base. (F– H) Immunoprecipitation of HA-tagged GFP, WT WDR60, WDR60 [T749M] and WDR60 [Q631*] mutant proteins followed by immunoblot for (F) HA, NudCD3, and GAPDH; red asterisks show expression of full-lengths WDR60 and truncated Q631*, (G) WDR34, LIC3/DYNC2LI1, and TCTEX1/DYNLT1 and (H) IFT140, IFT88, and IFT57.

DOI: https://doi.org/10.7554/eLife.39655.011

The following figure supplement is available for figure 7:

**Figure supplement 1.** Co-expression of disease-associated mutations do not affect IFT88 localization.
DOI: https://doi.org/10.7554/eLife.39655.012

WDR34 KO cells. This shows that the dynein-2 complex is largely intact in WDR34 KO cells. The only exception was DYNLRB1 which was found to bind to HA-WDR60 >1 $\log_2$-fold change less efficiently in WDR34 KO compared to WT cells. Furthermore, interaction of IFT-B and IFT-A with HA-WDR60 was only slightly reduced in the absence of WDR34. Notably, we found that several key components of the ciliary machinery, including the BBSome component, BBS7, and the IFT-B protein, IFT54, bound to WDR60 more tightly in the WDR34 KO cells than in WT cells. This suggests that loss of WDR34 leads to defects in axoneme extension by stalling assembled complexes and preventing dynamic assembly/disassembly of these vital IFT machineries.

## Discussion

### Structural and functional asymmetry of the dynein-2 motor

Our data show that the structural asymmetry within the dynein-2 motor is matched by functional asymmetry. Perhaps most strikingly, WDR34 is essential for axoneme extension during the early steps of ciliogenesis, whereas WDR60 is not required for cilium extension (the latter finding being validated recently by others [*Hamada et al., 2018*]). Depletion of WDR34 using RNAi is also associated with ciliary defects (*Asante et al., 2013*), whilst patient fibroblasts have shorter cilia with a bulbous tip (*Huber et al., 2013*) and fibroblasts from WDR34 knockout mice have stumpy cilia and defects in Shh signaling (*Wu et al., 2017*). It is intriguing that in our experiments some cells missing WDR34 can still extend a rudimentary cilium. However, even here, ciliary protein localization is severely disrupted. Thus, both subunits are necessary to maintain proper ciliary protein composition. Since WDR60 cells can extend an axoneme, WDR34 and WDR60 have clearly distinct but overlapping functions in cells.

Our proteomics and imaging data show that, in WDR34 KO cells, the system is stalled at the point of axoneme extension, after docking of the ciliary vesicle, despite apparently efficient co-assembly of the remaining dynein-2 motor and necessary IFT proteins. DHC2/DYNC2H1 and LIC3/DYNC2LI1 levels are, however, reduced at the ciliary base in WDR34 KO cells compared to WT cells. In contrast, several IFT proteins are shown to accumulate there. We suggest that in the absence of WDR34, dynein-2 can associate with the relevant large multimeric complex that includes IFT proteins and the BBSome, but cannot form the functional assemblies necessary for axoneme extension. One possibility is that WDR34 is required to target dynein-2 to the base of the nascent cilium, which is consistent with our immunofluorescence data. We did not identify any obvious components of the system that are missing in the absence of WDR34, suggesting that this is a mechanochemical defect. For example, the presence of WDR34 within the dynein-2 complex may induce a conformational change necessary for its concentration at the base of the cilium and to enable the functional assembly of IFT trains required for axoneme extension. Further structural work would be required to define this in molecular terms.

Paradoxically, our data also show that in the absence of WDR60, the dynein-2 holocomplex cannot form as effectively as it does in WT cells or in cells lacking WDR34, yet axoneme extension occurs normally. Unlike in WDR34 KO cells, we do observe DHC2/DYNC2H1 concentration at the

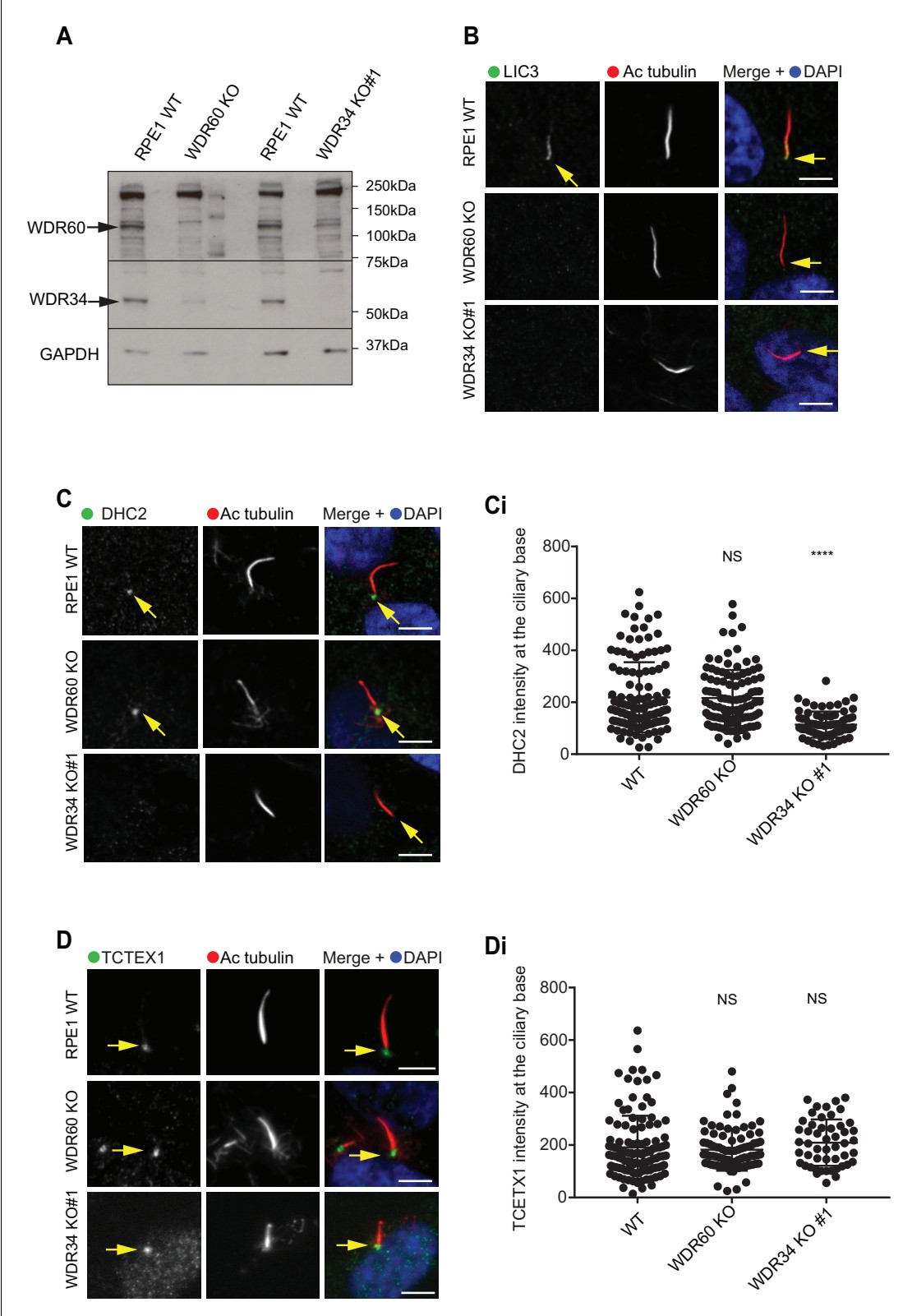

**Figure 8.** Dynein-2 assembly in primary cilium. (**A**) Immunoblotting for WDR60 and WDR34 in WT, WDR34 KO#1 and WDR60 KO cells. Arrows indicate WDR34 and WDR60 proteins. (**B**) LIC3/DYNC2LI1 localization in the cilia of WT, WDR34 KO#1 and WDR60 KO cells. (**C**) DHC2/DYNC2H1 localization at the ciliary base in WT and KO cells. (Ci) Intensity quantification shows a reduction of DHC2/DYNC2H1 at the ciliary base in WDR34 KO#1 cells (n = 3, 120 WT, 106 WDR60 KO, and 71 WDR34 KO #1 cells quantified). (**D**) TCTEX1/DYNLT1 localizes at the ciliary base in WT and KO cells. (Di) Intensity

*Figure 8 continued on next page*

*Figure 8 continued*

quantification of TCTEX1/DYNLT1 at the ciliary base (n = 3 115 WT, 85 WDR60 KO, and 50 WDR34 KO#1 cells quantified). Mann-Whitney test, p-value: ****=<0.0001. Scale bars 5 µm. Arrows point to the ciliary base.

DOI: https://doi.org/10.7554/eLife.39655.013

The following figure supplement is available for figure 8:

**Figure supplement 1.** Overexpression of WDR34 cannot rescue WDR60 KO phenotype and vice versa.

DOI: https://doi.org/10.7554/eLife.39655.014

base of the cilium in WDR60 KO cells, suggesting in this case targeting is not affected. Inefficient assembly of dynein-2 reduces interactions with IFT and BBS proteins but is sufficient and correctly positioned to permit axoneme extension. Thus, the WDR34-dependent, WDR60-independent, localization of DHC2/DYNC2H1 at the base of the cilium is required for axoneme extension. Meanwhile, reduced binding to IFT complexes and less efficient assembly of the functional dynein-2 holoenzyme both explain the IFT defects seen in WDR60 KO cells.

## Assembly of the dynein-2 holocomplex

Our data do not eliminate the possibility that WDR34 is itself required outside of the context of the dynein-2 complex. While we cannot rule out dynein-2-independent functions of WDR34 and WDR60, all of our data provide strong evidence that they co-exist in the dynein-2 holoenzyme. Interaction of WDR60 with WDR34 in our pull-down experiments indicates that dynein-2 complexes contain both WDR34 and WDR60, in agreement with our own previously reported data (*Asante et al., 2014*). Moreover, we found that stably expressed WDR60 could not rescue ciliogenesis defects in WDR34 KO cells, neither could WDR34 rescue IFT88 localization defects observed in WDR60 KO cells. Therefore WDR34 and WDR60 are not functionally redundant, at least in this regard. This supports a model where WDR34 and WDR60 play different roles in ciliogenesis and in IFT within the context of the dynein-2 complex, likely through different interactions with distinct components.

## Ciliary trafficking defects in absence of WDR34 and WDR60

Loss of either WDR34 or WDR60 leads to IFT particle accumulation at the base of as well as within cilia. We found that loss of WDR60 results in an increase of the IFT-B proteins, IFT20, IFT57, and IFT88, not only at the tip but also close to the base of the cilium. This suggests that IFT-B proteins could be retained at the basal body or around the transition zone. Consistent with these results, mutations in IFT-A or dynein-2 in mice also result in abnormal accumulation of IFT particles near the base of the cilium (*Goggolidou et al., 2014*; *Liem et al., 2012*; *Ocbina et al., 2011*). This has been linked to defects in the export of ciliary cargo across the transition zone (*He et al., 2017*). Similar defects are seen following disruption of the heavy chain, DHC2/DYNC2H1 (*Hou and Witman, 2015*).

In addition to these defects, we show that KAP3, a subunit of kinesin-2, accumulates at the ciliary tip of WDR60 KO cells. This would decrease the levels of kinesin-2 available to load onto departing anterograde trains and further cause accumulation of IFT particles at the base. This might also reflect some functional coupling of dynein-2 and kinesin-2 in the assembly of anterograde IFT particles. Notably, like WDR34 KO, KIF3B knockouts in both mice (*Nonaka et al., 1998*) and cultured cells (*Funabashi et al., 2018*) fail to form cilia. WDR34 might therefore be required to assemble functional IFT trains but we did not detect kinesin-2 in our WDR34 pull-down experiments. Our data are also consistent with models where, in metazoa, kinesin-2 motors are returned to the ciliary base by dynein-2-dependent retrograde IFT (*Broekhuis et al., 2013*; *Chien et al., 2017*; *Mijalkovic et al., 2017*; *Prevo et al., 2015*; *Signor et al., 1999*; *Williams et al., 2014*) but in contrast with *Chlamydomonas* studies showing kinesin-2 diffuses back to the ciliary base (*Engel et al., 2012*). Overall, our data support models where dynein-2 acts both in loading of cargo into cilia and in exit from cilia.

These findings suggest a complex interplay between IFT particles and IFT motors to control entry to and exit from cilia. Notably, interfering with either dynein-2 or IFT results in perturbation of transition zone organization, IFT, and ciliary membrane protein localization (*Garcia-Gonzalo and Reiter, 2017*).

# Loss of dynein-2 intermediate chains results in perturbed transition zone composition

The transition zone is the ciliary region most proximal to the mother centriole and it functions as a gate, acting as a diffusion barrier to prevent unregulated entry of high molecular weight proteins and maintaining the protein composition of the ciliary membrane (*Garcia-Gonzalo and Reiter, 2017*; *Jensen and Leroux, 2017*). Our results show that loss of either dynein-2 intermediate chain disrupts the organization of the ciliary transition zone. This could be significant in the context of Jeune syndrome where several cases result from mutations in WDR34 that are predicted to be complete loss of function mutations (e.g. c.472C > T (p.Gln158*) (*Schmidts et al., 2013b*)). The same is

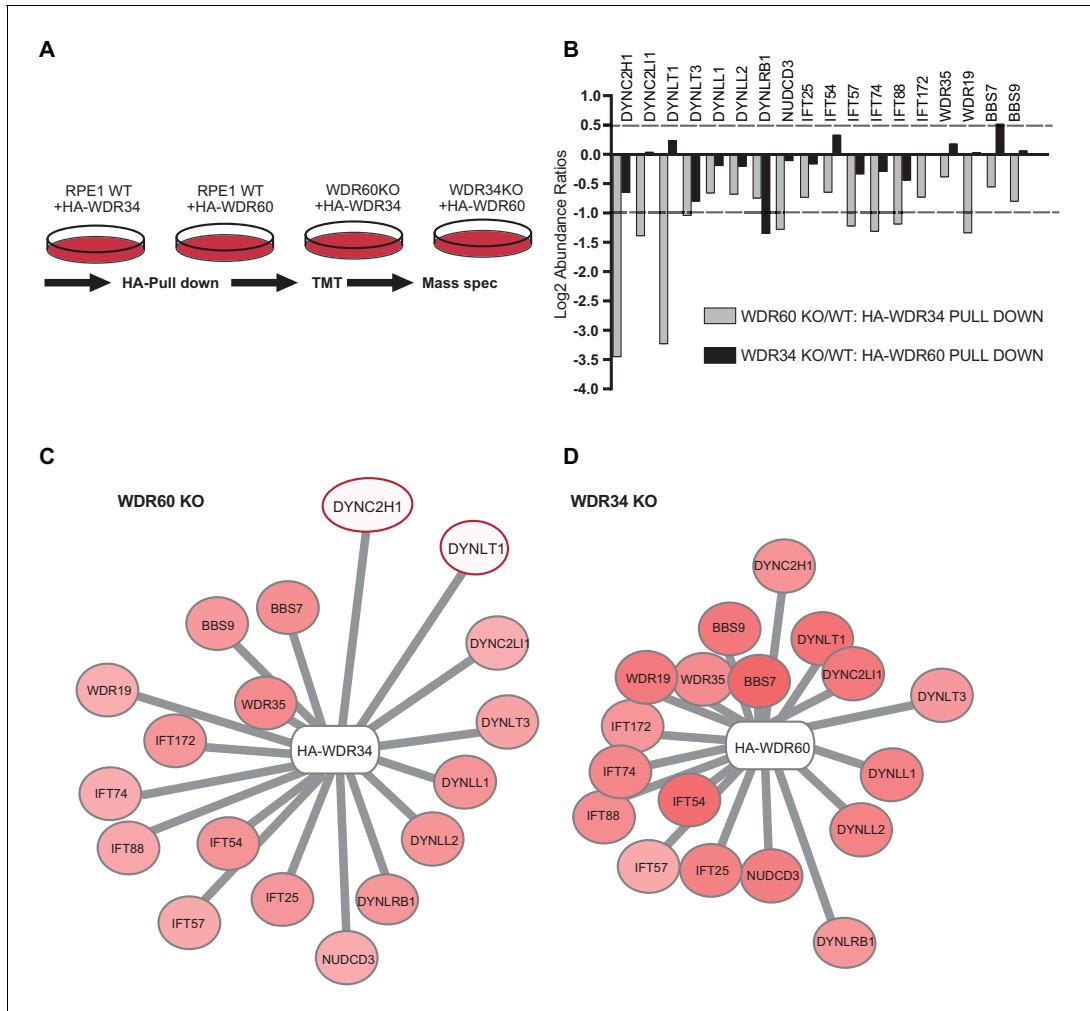

**Figure 9.** Proteomics of HA-WDR34 and HA-WDR60 interactomes in knockout cell lines. (**A**) Schematic representation of the HA pull downs and TMT proteomic methodology used. (**B**) The graph shows the log2-transformed ratios of the abundances between WDR60 KO/WT cells (grey) and WDR34KO/WT cells (black). Proteomic data were obtained from two independent experiments. The table shows raw data from one experiment. Similar results were obtained by normalizing the data with respect to the overexpressed protein abundance. (**C** and **D**) Schematic representations of HA-WDR34 and HA-WDR60 interactors with a red- white color scale. Red values correspond to an increased protein interaction and white to a reduced protein interaction with HA-WDR34 or HA-WDR60. The red circles indicate the two interactions most affected by the loss of WDR60. Distances represent the strength of protein-protein interaction in KO compared to WT cells. Protein more distant from the center present a decreased interaction with HA-WDR34/WDR60. Protein closer to the center present unchanged or increased interaction.

DOI: https://doi.org/10.7554/eLife.39655.015

The following figure supplement is available for figure 9:

**Figure supplement 1.** HA-WDR34 and HA-WDR60 expression in WT and KO cells.

DOI: https://doi.org/10.7554/eLife.39655.016

true for WDR60 (*McInerney-Leo et al., 2013*). However, no defects in transition zone structure have been described in cells derived from these patients. The defect in the transition zone could also explain the presence of intraciliary vesicles within the cilia of WDR60 KO cells that would normally be excluded by the diffusion barrier. The presence of intraciliary vesicles could also indicate a failure in the formation of ectosomes to remove the excess membrane from the cilia. Such vesicles have been described in motile cilia (*Shah et al., 2008*) and in mouse photoreceptor cells (*Gilliam et al., 2012*) from BBS mutants, as well as in wild-type zebrafish (*Goetz et al., 2014*). However, we always see swollen ciliary tips in WDR60 KO cells when imaging living cells expressing ciliary membrane markers and have not detected any shedding of vesicles during such experiments. This suggests that any mechanisms to reduce membrane accumulation might not be able to overcome any defect in retrograde IFT.

The role of dynein-2 in maintaining transition zone composition is also consistent with the fact that LIC3/DYNC2LI1 and overexpressed WDR34 and WDR60 localize at the transition zone in RPE1 cells. Moreover, in a previous study, active phosphorylated TCTEX1/DYNLT1 was detected at the transition zone of neural progenitors (*Li et al., 2011*). Changes in transition zone composition are associated with a reduced localization of soluble and membrane proteins in the cilium (*Berbari et al., 2008b*; *Chih et al., 2011*). 5HT6, SSTR3, and Arl13B are all reduced in abundance within cilia of WDR60 KO cells, suggesting that these proteins might not be effectively retained within cilia, but leak out through the diffusion barrier. An alternative possibility is that these proteins are less effectively loaded into cilia. We did not find a difference in the intensity levels of overexpressed Rab8a in WDR60 KO compared to WT cells suggesting that at least some proteins can enter normally, likely reflecting a differential requirement for dynein-2 function. Our data do not discriminate between direct or indirect roles for dynein-2 in either building or maintaining the ciliary transition zone. Overall, our data suggest that the reduced interaction of dynein-2 with IFT and BBS proteins likely underpins these defects in the transition zone structure in WDR60 KO cells. Indeed, it is quite likely that the role of dynein-2 in transition zone structure is a result of its established role in IFT. Recent data from the Blacque lab using IFT-A mutants (*Scheidel and Blacque, 2018*) and from the Leroux labs using a temperature sensitive allele of the dynein-2 heavy chain (Jensen et al., *personal communication*), both using *C. elegans*, provides support for this interpretation.

## Dynein-2 dysfunction and disease

The list of mutations causing disease in genes associated with primary cilia is continuously expanding. Our results show that Smo localization is deregulated in the absence of WDR60. Smo is thought to enter cilia continuously but then rapidly exits in the absence of ligand. Our data are consistent with this, with loss of WDR60 resulting in aberrant accumulation of Smo in cilia, likely leading to perturbed hedgehog signaling (also recently validated by others [*Hamada et al., 2018*]). Interestingly, abnormal accumulation of Smo in the cilia has also been observed in mouse fibroblasts with mutations in DHC2/DYNC2H1. In these mutant mice, inactivation of dynein-2 causes loss of Shh signaling and midgestation lethality (*Ocbina et al., 2011*). Given the links between hedgehog signaling and skeletogenesis, this is likely to be a major cause of the phenotypes seen on loss-of-function of dynein-2 in animal models and in patients (*Dagoneau et al., 2009*; *Li et al., 2015*; *May et al., 2005*; *Wu et al., 2017*).

In this study, we also characterized the function of dynein-2 using disease-causing mutations found in SRPs and JATD syndromes (*McInerney-Leo et al., 2013*). Using mutagenesis to recreate patient mutations WDR60[Q631*] and WDR60[T749M], we show that the N-terminal region of WDR60 is sufficient to bind to WDR34 and TCTEX1/DYNLT1 but not to LIC3/DYNC2LI1. Further analysis of patient mutation [Q631*] reveals that the C-terminal β-propeller domain of WDR60 is required for binding of the IFT-B proteins, including IFT88. The data do not let us conclude whether there is a direct interaction between WDR60 and IFT-B or whether WDR60 is required to assemble intact dynein which in turn can bind to IFT-B.

The observed reduction in binding between the WDR60[Q631*] mutant and NudCD3 is expected as the NudC family act as co-chaperones with Hsp90 to fold β-propellers such as WD repeat domains (*Taipale et al., 2014*). In contrast, WDR60[T749M] binds to other dynein-2 proteins, IFT proteins, and NudCD3. The reduction in binding to LIC3/DYNC2LI1 seen with this mutation suggests that less efficient dynein-2 assembly might contribute to the patient phenotypes. While these

biochemical data are clear, a caveat here is that we are overexpressing these mutants which could overcome subtle defects.

Our data show that not only is dynein-2 required for retrograde IFT but also to build and maintain a functional diffusion barrier at the base of the cilium. Our data do not discriminate between roles in assembly versus maintenance of the transition zone. Other recent work has shown that Joubert syndrome is caused by disruption of the transition zone (*Shi et al., 2017*). Joubert syndrome leads to severe neurological effects that are underpinned by developmental defects in hedgehog signaling. Jeune syndrome is also attributed to disrupted hedgehog signaling but in this case this manifests as skeletal defects. Together our work and that of *Shi et al., 2017* suggests that disruption of transition zone architecture and resulting defects in developmental signaling might define a common root cause of both Joubert and Jeune syndromes, and indeed perhaps other ciliopathies.

# Materials and methods

## Key resources table

| Reagent type (species) or resource | Designation | Source or reference | Identifiers | Additional information |
|---|---|---|---|---|
| Cell line (*Homo sapiens*) | HEK293T | ATCC CRL-3216 | RRID:CVCL_0063 | Purchased from ATCC, not verified further |
| Cell line (*Homo sapiens*) | hTERT-RPE-1 | ATCC CRL-4000 | RRID:CVCL_4388 | Purchased from ATCC, not verified further |
| Antibody | Acetylated tubulin | Sigma-Aldrich | Cat# T6793 RRID:AB_477585 | 1:2000 for IF |
| Antibody | HA | Cell Signaling Technology | Cat# 3724 RRID:AB_1549585 | 1:2000 WB, 1:1000 IF |
| Antibody | IFT88 | Proteintech Group | Cat# 13967–1-AP RRID:AB_2121979 | 1:200 WB, 1:300 IF |
| Antibody | IFT140 | Proteintech Group | Cat# 17460–1-AP RRID:AB_2295648 | 1:200, WB 1:100 IF |
| Antibody | IFT43 | Proteintech Group | Cat# 24338–1-AP RRID:AB_2749824 | 1:50 IF |
| Antibody | IFT20 | Proteintech Group | Cat# 13615–1-AP RRID:AB_2280001 | 1:200 IF |
| Antibody | TMEM67 | Proteintech Group | Cat# 13975–1-AP RRID:AB_10638441 | 1:50 IF |
| Antibody | RPGRIP1L | Proteintech Group | Cat# 55160–1-AP RRID:AB_10860269 | 1:100 IF |
| Antibody | DYNC2H1 | Proteintech Group | Cat# 55473–1-AP AB_2749823 | 1:100 IF |
| Antibody | DYNC2LI1/LIC3 | Proteintech Group | Cat# 15949–1-AP RRID:AB_2093643 | 1:250 WB, 1:100 IF |
| Antibody | TCTEX1 | Santa Cruz Biotechnology | Cat# sc-28537 RRID:AB_2093671 | 1:200 WB, 1:100 IF |
| Antibody | Arl13B | Proteintech Group | Cat# 17711–1-AP RRID:AB_2060867 | 1:1000 IF |
| Antibody | TCTN1 | Proteintech Group | Cat# 15004–1-AP RRID:AB_10644442 | 1:100 IF |
| Antibody | Smoothened | Abcam | Cat# ab38686 RRID:AB_882615 | 1:100 IF |
| Antibody | Myc | gift from Harry Mellor | PMID:20233848 | 1:1000 IF |
| Antibody | WDR60 | Sigma-Aldrich | Cat# HPA021316 RRID:AB_2215577 | 1:300 WB in *Figure 1—figure supplement 2* and *Figure 9—figure supplement 1* |

*Continued on next page*

*Continued*

| Reagent type (species) or resource | Designation | Source or reference | Identifiers | Additional information |
|---|---|---|---|---|
| Antibody | WDR60 | Novus | Cat# NBP1-90437 RRID:AB_11023602 | WB in *Figure 9* |
| Antibody | WDR34 | Novus | Cat# NBP1-88805 RRID:AB_11006071 | 1:300 WB |
| Antibody | GAPDH | Abcam | Cat# ab9484 RRID:AB_307274 | 1:1000 WB |
| Antibody | p150glued | BD Labs | Cat# 612709 RRID:AB_399948 | 1:1000 WB |
| Antibody | LIS1 | Bethyl | Cat# A300-409A RRID:AB_2159907 | 1:1000 WB |
| Antibody | dic74 | Millipore | Cat# MAB1618 RRID:AB_2246059 | 1:1000 WB |
| Antibody | NUDCD3 | Atlas Antibodies | Cat# HPA019136 RRID:AB_1852370 | 1:350 WB |
| Antibody | Gamma-Tubulin | Sigma-Aldrich | Cat# T5326, RRID:AB_532292 | |
| Antibody | Anti-HA-Agarose | Sigma-Aldrich | Cat# A2095 RRID:AB_257974 | |
| Antibody | Alexa- secondary antibodies | Invitrogen | | 1:1000 |
| Antibody | HRP-secondaries | Jackson ImmunoResearch | | |
| recombinant DNA reagent | pSpCas9(BB)—2A-GFP | Addgene | Cat# PX458 PMID:24157548 | |
| recombinant DNA reagent | pGEM T Easy vector | PROMEGA | Cat# A1360 | |
| recombinant DNA reagent | L13-Arl13b-GFP (plasmid) | gift from Tamara Caspary/Addgene | Cat# 40879 PMID:21976698 | |
| recombinant DNA reagent | IFT20-GFP (plasmid) | gift from Gregory Pazour/Addgene | Cat# 45608 PMID:16775004 | |
| recombinant DNA reagent | pEGFPN3-SSTR3 (plasmid) | gifts from Kirk Mykytyn/Addgene | Cat# 35624 PMID:18256283 | |
| recombinant DNA reagent | pEGFPN3-5HT6 (plasmid) | Addgene | Cat# 35623 PMID:18256283 | |
| recombinant DNA reagent | EGFP-Rab8 (plasmid) | gift from Johan Peränen | | |
| recombinant DNA reagent | pLVX-Puro-mGFP -WDR34 (plasmid) | PMID:25205765 | NM_052844.3, NP_443076 | |
| recombinant DNA reagent | WDR60 (cDNA) | Life Technologies | Uniprot: Q8WVS4 ENSG00000126870 | |
| recombinant DNA reagent | pLVX-Puro-HA-WDR60 (plasmid) | this paper | | |
| recombinant DNA reagent | pLVX-Puro-HA-WDR60 T749M (plasmid) | this paper | | |
| recombinant DNA reagent | pLVX-Puro-Myc-WDR60 T749M (plasmid) | this paper | | |
| recombinant DNA reagent | pLVX-Puro-HA-WDR60 Q631* (plasmid) | this paper | | |
| chemical compound, drug | SAG | Selleckchem | Cat# S7779 | |

*Continued on next page*

*Continued*

| Reagent type (species) or resource | Designation | Source or reference | Identifiers | Additional information |
|---|---|---|---|---|
| chemical compound, drug | DAPI | Life Technologies | Cat# D1306 | |
| chemical compound, drug | Amersham ECL | GE Healthcare | Cat# RPN2106 | |
| chemical compound, drug | DSP | Thermo Fisher Scientific | Cat# 22585 | |
| chemical compound, drug | Protease inhibitors | Millipore | Cat# 539137 | |
| chemical compound, drug | LDS sample buffer | Life Technologies | Cat# NP007 | |
| chemical compound, drug | Sample reducing agent | Life Technologies | Cat# NP007 | |
| chemical compound, drug | Lipofectamine 2000 | Life Technologies | Cat# 11668027 | |
| chemical compound, drug | Lenti-XTM HTX Packaging System | Clontech | Cat# 631275 | |
| Sequence-based reagent | WDR34 gRNA sequences (Exon2) | 5'-AGCCTTTCTTC GGAGAGTGG-3' | | |
| Sequence-based reagent | WDR34 gRNA sequences (Exon3) | 5'-CAGGTGTCTTGT CTGTATAC −3' | | |
| Sequence-based reagent | WDR60 gRNA sequences (Exon3) | 5'-AGGTGCAGGGA TCCCGACCA-3' | | |
| Software and Algorithms | FIJI/ImageJ | https://fiji.sc/ | PMID:22743772 PMID:22743772 PMID:22743772 | |
| Software and Algorithms | Prism 6 | http://www.graphpad.com | | |
| Software and Algorithms | Proteome Discoverer software v2.1 | Thermo Fisher Scientific | | |

All reagents were purchased from Sigma-Aldrich (Poole, UK) unless stated otherwise

## Plasmids, cloning, and mutagenesis

The human WDR34 gene was obtained from the Origene (SC319901, Cambridge Bioscience), human WDR60 was generated by gene synthesis (Life Technologies, Paisley, UK). An HA tag or Myc tag for WDR34 and WDR60 was added by PCR and both proteins were subcloned in the pLVX-puro vector. Mutant [T749M WDR60] was generated by site-directed mutagenesis PCR using the primers: Fw: 5'-CAGAACCGCCatgTTCTCCACC-3' and Rv 5'-GGTGGAGAACATGGCGGTTCTG-3' changing codon ACG (Threonine) to ATG (Methionine). WDR60 [Q631*] mutant was constructed by site-directed mutagenesis using the primers: Fw: 5'-GATAGCAGCTCCtagCTGAATACC-3' and 5'-GGTATTCAGC TAGGAGCTGCTATC-3' changing codon CAG (Glutamine) to TAG (STOP codon). All constructs were validated by DNA sequencing.

Mouse L13-Arl13b-GFP was a gift from Tamara Caspary (Addgene plasmid # 40879, [*Larkins et al., 2011*]), IFT20-GFP ([*Follit et al., 2006*], plasmid JAF2.13) was a gift from Gregory Pazour (Addgene plasmid # 45608). pEGFPN3-SSTR3 and pEGFPN3-5HT6 were gifts from Kirk Mykytyn (Addgene plasmid #35624 and #35623, [*Berbari et al., 2008a*]), EGFP-Rab8 was a gift from Johan Peränen (University of Helsinki).

## Cell culture

Cell lines were purchased from ATCC and not verified further other than to confirm no mycoplasma contamination. Human telomerase-immortalized retinal pigment epithelial cells (hTERT-RPE-1, ATCC CRL-4000) were grown in DMEM-F12 supplemented with 10% FBS (Life Technologies, Paisley, UK) at 37°C under 5% $CO_2$. Cells were not validated further after purchase from ATCC. Transient

transfections of Arl13b-GFP, SSTR3-GFP, 5HT6-GFP, and Rab8-GFP were performed using Lipofect-amine 2000 (Life Technologies, Paisley, UK) according to the manufacturer's protocol. Lentiviral particles for each of the stable RPE-1 cell lines were produced in HEK293T cells using the Lenti-XTM HTX Packaging System (Clontech, Saint-Germain-en-Laye, France). Low passage hTERT-RPE1 cells were transduced with the resultant viral supernatant, strictly according to the manufacturer's directives and at 48 hr post-transduction, cells were subcultured in presence of 5 µg/ml puromycin. RPE-1 cells were incubated in serum-free medium for 24 hr to induce ciliogenesis. Confluent cells were placed in serum-free media and treated with Shh agonist SAG (Selleckchem (from Stratech Scientific, Ely, UK) Catalog No.S7779) at the final concentration of 100 nM for 24 hr.

## Genome engineering

The guide RNAs (gRNA) targeting WDR34 were designed using 'chop chop' software (*Labun et al., 2016*) or using CRISPR design http://crispr.mit.edu/ for designing WDR60 gRNA (*Hsu et al., 2014*). pSpCas9(BB)−2A-GFP (Addgene plasmid, #PX458) was used as the vector to generate a gRNA. The gRNA sequences (5'- A GCC TTT CTT CGG AGA GTG G-3'; and 5'-CA GGT GTC TTG TCT GTA TAC −3') were designed to target Exon2 and Exon3 of human WDR34. Similarly, the gRNA (5'-AG GTG CAG GGA TCC CGA CCA-3') was designed to target exon 3 of WDR60. RPE-1 cells were transfected with 1 µg of pSpCas9(BB)−2A-GFP. After 48 hr GFP-positive cells were sorted, and singles cells were plated in a 96 well plate. To check the WDR34 and WDR60 genes, genomic DNA was extracted and the target sequences subjected to PCR. Subsequently, the PCR products were cloned in the pGEM T Easy vector according to the manufacturer's instructions and sequenced. In three cells clones, identified as WDR34 KO#1, WDR34 KO#2 and WDR60 KO, small deletions/insertions causing a frameshift were detected in both alleles (Supplementary *Figure 1* for details). On the contrary, the cell clones identified as CRISPR CTRL WDR34 and WDR60 cells, transfected and treated in the same conditions of our knock out clones, did not show any mutation in the targeted genomic DNA region.

## Antibodies

The antibodies used, and their dilutions for western blotting (WB) and immunofluorescence (IF) are as follows: Acetylated tubulin (Sigma (Poole, UK) T6793 1:2000 for IF), rabbit anti-HA (Cell Signaling Technologies (New England Biolabs, Hitchin, UK) 1:2000 WB, 1:1000 IF), rabbit IFT88 (Proteintech (Manchester, UK) 13967−1-AP, 1:200 WB, 1:300 IF), rabbit anti-IFT140 (Proteintech 17460−1-AP, 1:200, WB 1:100 IF), rabbit anti-IFT57 (Proteintech 11083−1,-AP 1:200, WB 1:100 IF), rabbit anti-IFT43 (Proteintech 24338−1-AP, 1:50 IF), anti-IFT20 (Proteintech 13615−1-AP, 1:200 IF), anti-TMEM67 (proteintech13975-1-AP, 1:50 IF), anti-RPGRIP1L (Proteintech 55160−1-AP, 1:100 IF), anti-DYNC2HC1 (Proteintech 55473−1-AP, 1:100 IF), rabbit anti-LIC3 (Proteintech 15949−1-AP, 1:250 WB, 1:100 IF), rabbit anti-TCTEX1 (Santa Cruz Biotechnology (from Insight Biotechnology, Wembley, UK) sc-28537, 1:200 WB, 1:100 IF), rabbit anti-Arl13B (Proteintech 17711-1AP, 1:1000 IF), rabbit anti-TCTN1 (Proteintech 15004−1-AP, 1:100 IF), rabbit anti-Smo (Abcam (Cambridge, UK) ab38686, 1:100 IF), Sheep anti-Myc ([*Fan et al., 2010*] kindly provided by Harry Mellor, University of Bristol), rabbit anti-WDR60 (Novus Biologicals (from Bio-Techne Abingdon, UK) NBP1-90437 1:300 WB in *Figure 9*), rabbit anti-WDR60 (Sigma HPA021316, 1:300 WB in *Figure 1—figure supplement 2* and *Figure 9—figure supplement 1*), rabbit anti-WDR34 (Novus NBP188805, 1:300 WB), mouse anti-GAPDH (Abcam ab9484, 1:1000 WB), p150 glued (BD 6127009, city 1:1000 WB), LIS1 (Bethyl A300-409A, (Montgomery, TX) 1:1000 WB), dic74 (MAB1618 Millipore (Watford, UK), 1:1000 WB), NUDCD3 (Sigma HPA019136, 1:350 WB).

## Immunofluorescence

Cells grown on 0.17 mm thick (#1.5) coverslips (Fisher Scientific, Loughborough, UK) were washed in PBS, and then fixed for 10 min in PFA and permeabilized with PBS containing 0.1% Triton X-100 for 5 min. Alternatively, cells were fixed in ice-cold methanol at −20°C for 5 min for TMEM67, RPGRIP1L, TCTN1, DYNC2H1 and IFT20 staining.

For TCTEX1 immunolabelling, cells were washed twice with pre-warmed cytoskeletal buffer (CB, containing 100 mM NaCl 300 mM sucrose, 3 mM MgCl2, and 10 mM PIPES) and fixed for 10 min in CB-PFA, as described previously (*Hua and Ferland, 2017*). Subsequently, cells were blocked using

3% BSA in PBS for 30 min at room temperature. The coverslips were incubated with primary antibodies for 1 hr at room temperature, washed in PBS and then incubated with secondary antibodies for another 1 hr at room temperature. Nuclear staining was performed using DAPI [4,6-diamidino-2-phenylindole (Life Technologies), diluted at 1:5000 in PBS] for 3 min at room temperature, and the cells were then rinsed twice in PBS. Cells were imaged using an Olympus IX- 71 or IX-81 widefield microscope with a 63x objective, and excitation and emission filter sets (Semrock, Rochester, NY) controlled by Volocity software (version 4.3, Perkin-Elmer, Seer Green, UK). Alternatively, cells in *Figure 4* and *Figure 7* were imaged using Leica SP5 confocal microscope (Leica Microsystems, Milton Keynes, UK). Live images in *Figure 3* were imaged using Leica SP8. All images were acquired as 0.5 µm z-stacks. All graphs show mean and standard deviation.

### Rescue experiments

For 'rescue' experiments, stable WDR60 KO cell lines overexpressing wild-type and HA-tagged WDR60 mutants were generated. Cells were serum starved for 24 hr, fixed and processed for immunofluorescence analysis.

### Electron microscopy

Cells were serum starved 24 hr and fixed in 2.5% glutaraldehyde for 20 min. Next, the cells were washed for 5 min in 0.1 M cacodylate buffer then post-fixed in 1% $OsO_4$/0.1 M cacodylate buffer for 30 min. Cells were washed 3x with water and stained with 3% uranyl acetate for 20 min. After another rinse with water, cells were dehydrated by sequential 10 min incubations with 70, 80, 90, 96, 100% and 100% ethanol before embedding in Epon at 70°C for 48 hr. Thin (70 nm) serial sections were cut and stained with 3% uranyl acetate then lead citrate, washing 3x with water after each. Once dried, sections were imaged using an FEI (Cambridge, UK) Tecnai12 transmission electron microscope.

### Immunoblotting

Cells were lysed in buffer containing 50 mM Tris pH7.5, 150 mM NaCl, 1% Igepal and 1 mM EDTA. Samples were separated by SDS-PAGE followed by transfer to nitrocellulose membranes. Membranes were blocked in 5% milk-TBST. Primary antibodies diluted in blocking buffer were incubated with membrane overnight and detected using HRP-conjugated secondary antibodies (Jackson ImmunoResearch, West Grove, PA) and enhanced chemiluminescence (GE Healthcare, Cardiff, United Kingdom).

### Fluorescence intensity measurement

Quantification of fluorescence intensity was performed using original images. Measurement of intensity was performed using the average projections of acquired z-stacks of the area of the ciliary marker acetylated tubulin. Fluorescence intensity along the ciliary axoneme was measured using ImageJ plot profile tool. Fluorescence intensity at the ciliary base was measured drawing same diameter circles at the ciliary base.

### Immunoprecipitation

RPE-1 cells expressing the indicated cDNA constructs were washed with PBS and incubated with crosslinker solution (1 mM DSP, Thermo Fisher Scientific #22585) for 30 min on ice. The reaction was quenched by adding 500 mM Tris-HCl pH 7.5 for 15 min. Cells were washed twice with PBS and lysed in a buffer containing 50 mM Tris/HCl, pH 7.4, 1 mM EDTA, 150 mM NaCl, 1% Igepal and protease inhibitors (539137, Millipore). Subsequently, cells were incubated on a rotor at 4 °C for 30 min and then lysates were centrifuged at 13,000 g at 4 °C for 10 min. Cell lysates were added to the equilibrated anti-HA-Agarose beads (Sigma A2095, batch number 026M4810V) and incubated on a rotor at 4 °C. Next, the beads were washed three times by centrifuging at 2000 g for 2 min at 4 °C with 1 ml of washing buffer (50 mM Tris-HCl, 150 mM NaCl, 0.5 mM EDTA, Triton X-100 0.3% SDS 0.1%) containing protease inhibitors (539137, Millipore). Samples used for SDS-PAGE and immunoblotting were resuspended in 50 µl of LDS sample buffer (NP007, Life Technologies) containing sample reducing agent (NP007, Life Technologies) and boiled at 95 °C for 10 min.

## Proteomic analysis

For TMT Labelling and high pH reversed-phase chromatography, the samples were digested from the beads with trypsin (2.5 μg trypsin, 37°C overnight), labeled with Tandem Mass Tag (TMT) six-plex reagents according to the manufacturer's protocol (Thermo Fisher Scientific, Loughborough, UK) and the labeled samples pooled. The pooled sample was then desalted using a SepPak cartridge according to the manufacturer's instructions (Waters, Milford, Massachusetts, USA)). Eluate from the SepPak cartridge was evaporated to dryness and resuspended in buffer A (20 mM ammonium hydroxide, pH 10) prior to fractionation by high pH reversed-phase chromatography using an Ultimate 3000 liquid chromatography system (Thermo Fisher Scientific). In brief, the sample was loaded onto an XBridge BEH C18 Column (130 Å, 3.5 μm, 2.1 mm X 150 mm, Waters, UK) in buffer A and peptides eluted with an increasing gradient of buffer B (20 mM ammonium hydroxide in acetonitrile, pH 10) from 0–95% over 60 min. The resulting fractions were evaporated to dryness and resuspended in 1% formic acid prior to analysis by nano-LC MSMS using an Orbitrap Fusion Tribrid mass spectrometer (Thermo Fisher Scientific).

## Nano-LC mass spectrometry

High pH RP fractions were further fractionated using an Ultimate 3000 nano-LC system in line with an Orbitrap Fusion Tribrid mass spectrometer (Thermo Fisher Scientific). In brief, peptides in 1% (vol/vol) formic acid were injected onto an Acclaim PepMap C18 nano-trap column (Thermo Fisher Scientific). After washing with 0.5% (vol/vol) acetonitrile 0.1% (vol/vol) formic acid, peptides were resolved on a 250 mm $\times$75 μm Acclaim PepMap C18 reverse phase analytical column (Thermo Fisher Scientific) over a 150 min organic gradient, using seven gradient segments (1–6% solvent B over 1 min., 6–15% B over 58 min., 15–32% B over 58 min., 32–40% B over 5 min., 40–90% B over 1 min., held at 90% B for 6 min and then reduced to 1% B over 1 min.) with a flow rate of 300 nl min$-$1. Solvent A was 0.1% formic acid and Solvent B was aqueous 80% acetonitrile in 0.1% formic acid. Peptides were ionized by nano-electrospray ionization at 2.0 kV using a stainless steel emitter with an internal diameter of 30 μm (Thermo Fisher Scientific) and a capillary temperature of 275°C.

All spectra were acquired using an Orbitrap Fusion Tribrid mass spectrometer controlled by Xcalibur 2.0 software (Thermo Fisher Scientific) and operated in data-dependent acquisition mode using an SPS-MS3 workflow. FTMS1 spectra were collected at a resolution of 120 000, with an automatic gain control (AGC) target of 400 000 and a max injection time of 100 ms. Precursors were filtered with an intensity range from 5000 to 1E20, according to charge state (to include charge states 2–6) and with monoisotopic precursor selection. Previously interrogated precursors were excluded using a dynamic window (60 s $\pm$ 10 ppm). The MS2 precursors were isolated with a quadrupole mass filter set to a width of 1.2 m/z. ITMS2 spectra were collected with an AGC target of 10 000, max injection time of 70 ms and CID collision energy of 35%.

For FTMS3 analysis, the Orbitrap was operated at 30 000 resolution with an AGC target of 50 000 and a max injection time of 105 ms. Precursors were fragmented by high energy collision dissociation (HCD) at a normalized collision energy of 55% to ensure maximal TMT reporter ion yield. Synchronous Precursor Selection (SPS) was enabled to include up to 5 MS2 fragment ions in the FTMS3 scan.

## Data analysis

The raw data files were processed and quantified using Proteome Discoverer software v2.1 (Thermo Fisher Scientific) and searched against the UniProt Human database (140000 entries) and GFP sequence using the SEQUEST algorithm. Peptide precursor mass tolerance was set at 10 ppm, and MS/MS tolerance was set at 0.6 Da. Search criteria included oxidation of methionine (+15.9949) as a variable modification and carbamidomethylation of cysteine (+57.0214) and the addition of the TMT mass tag (+229.163) to peptide N-termini and lysine as fixed modifications. Searches were performed with full tryptic digestion and a maximum of 1 missed cleavage was allowed. The reverse database search option was enabled and the data was filtered to satisfy false discovery rate (FDR) of 5%.

## Acknowledgements

We would like to thank Katharine Risk, Beth Moyse, and Imogen Binnian for their contributions to the work and Janine McCaughey for helpful discussion on this work. Thanks also to Max Nachury and Jackie Goetz for helpful comments. This work was supported by grants from the BBSRC (BB/N000420/1) and MRC (MR/P000177/1 and MR/K018019/1). We would like to thank the Wolfson Bioimaging Facility for support of the microscopy experiments. Confocal microscopy was supported by a BBSRC capital grant (BB/L014181/1).

## Additional information

### Funding

| Funder | Grant reference number | Author |
| --- | --- | --- |
| Medical Research Council | MR/P000177/1 | Nicola L Stevenson<br>David John Stephens |
| Biotechnology and Biological Sciences Research Council | BB/N000420/1 | Laura Vuolo<br>David John Stephens |
| Medical Research Council | MR/K018019/1 | Nicola L Stevenson<br>David John Stephens |
| Biotechnology and Biological Sciences Research Council | BB/L014181/1 | David John Stephens |

The funders had no role in study design, data collection and interpretation, or the decision to submit the work for publication.

### Author contributions

Laura Vuolo, Conceptualization, Data curation, Formal analysis, Validation, Investigation, Visualization, Methodology, Writing—original draft, Writing—review and editing; Nicola L Stevenson, Conceptualization, Resources, Data curation, Formal analysis, Validation, Investigation, Visualization, Methodology, Writing—original draft, Writing—review and editing; Kate J Heesom, Formal analysis, Investigation, Methodology, Writing—original draft; David J Stephens, Conceptualization, Resources, Data curation, Formal analysis, Supervision, Funding acquisition, Validation, Investigation, Visualization, Methodology, Writing—original draft, Project administration, Writing—review and editing

### Author ORCIDs

Laura Vuolo http://orcid.org/0000-0002-9801-9206
Nicola L Stevenson http://orcid.org/0000-0001-8967-7277
Kate J Heesom http://orcid.org/0000-0002-5418-5392
David J Stephens http://orcid.org/0000-0001-5297-3240

### Decision letter and Author response

Decision letter https://doi.org/10.7554/eLife.39655.023
Author response https://doi.org/10.7554/eLife.39655.024

## Additional files

### Supplementary files

• Supplementary file 1. Excel file listing proteomic results. Page one shows results of HA-WDR60 pull down in WT and WDR34 KO cells. Page two shows results of HA-WDR34 pull down in WT and WDR60 KO cells. 'Description' refers to *Homo sapiens* RefSeq database. 'Identified in # experiments' shows protein presence in 1/2 or 2/2 separate analyses. Proteins are listed in order of abundance comparing WDR34 or WDR60 KO cells respect to WT.
DOI: https://doi.org/10.7554/eLife.39655.017

• Transparent reporting form
DOI: https://doi.org/10.7554/eLife.39655.018

### Data availability

Proteomics data are included as an Excel file which is readily readable and accessible for the majority of researchers. The mass spectrometry proteomics data have been deposited to the ProteomeX-change Consortium via the PRIDE [1] partner repository with the dataset identifier PXD010398.

The following dataset was generated:

| Author(s) | Year | Dataset title | Dataset URL | Database and Identifier |
|---|---|---|---|---|
| Laura Vuolo, Nicola L Stevenson, Kate J Heesom, David John Stephens | 2018 | Mass spectrometry proteomics data | https://www.ebi.ac.uk/pride/archive/projects/PXD010398 | EBI PRIDE, PXD010398 |

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
