## [Decision Letter]

[Editors’ note: a previous version of this study was rejected after peer review, but the authors submitted for reconsideration. The first decision letter after peer review is shown below.]

Thank you for submitting your work entitled "Dynein-2 intermediate chains play crucial but distinct roles in primary cilia formation and function" for consideration by *eLife*. Your article has been reviewed by three peer reviewers, and the evaluation has been overseen by a Reviewing Editor and a Senior Editor. The reviewers have opted to remain anonymous.

The reviewers found the manuscript contained a number of interesting findings but after discussion felt that there was not currently enough mechanistic insight to warrant publication in *eLife*. We therefore regret to that we are rejecting the paper at this stage.

To make your manuscript appropriate for e*Life* the reviewers would want to see mechanistic insight into one or both of the following observations:

1) One of the highlights of your paper is the finding that WDR34 knockout inhibits ciliogenesis. Both reviewers 2 and 3 suggest following up on mass spec experiments to identify what factors bind WDR34 in order to put forward some mechanism.

2) Another highlight is the fascinating observation that dynein-2 IC mutations interfere with the transition zone at the base of the cilia. How do they do this? For example does loss of other components of the IFT particle or IFT motors generate similar defects (see reviewer 2)?

The other major point that the reviewers would need to be addressed to consider any re-submission would be the mass spec experiments (reviewer 3 – point 17). I include all the reviewer's comments below.

*Reviewer #1:*

Cilia assembly, maintenance, and functioning is dependent on bidirectional intraflagellar transport between its base and growing tip. Transport back to the base is carried out by cytoplasmic dynein-2 that unlike cytoplasmic dynein-1 has an asymmetric tail complex. The two different intermediate chains (ICs) WDR34 and WDR60 in this tail have been linked to cilia related disease. However a molecular characterization of the function of the asymmetric tail subunits remained absent.

The authors show in human cells that depletion and patient mutations in both WDR60 and WDR34 lead to overlapping disturbances in ciliary membrane composition and transition zone integrity. Depletion of WDR60 disrupts the complete dynein complex, affecting retrograde transport, while cilia are still able to form. In contrast, WDR34 depletion does not disrupt the full dynein complex yet ciliogenesis is severely affected. With this the authors show that the different subunits have overlapping but also distinct roles that are not limited to retrograde transport.

The findings in this manuscript are interesting and the data is solid. However, the absence of mechanistic data leaves this reviewer to contemplate whether it is suited for publication in e*Life*.

1) The authors clearly show that the transition zone and membrane composition of the cilium are perturbed in relation to IC depletion. However, there is no mechanistic data for how the transition zone is perturbed. This leaves us with the question how far downstream the transition zone perturbation is from dynein perturbation. I would like to see how dynein depletion leads to disruption of the transition zone.

2) It is striking that, according to the MS data, the unassembled [WDR60 KO] dynein complex leads to a less severe phenotype than the partially intact [WDR34 KO] dynein complex. The partially assembled dynein that lacks WDR34 results in a ten-fold reduction in ciliated cells. In the MS data, HA-WDR34 pulls down all components less well in the [WDR60 KO] than the WT, showing the dynein complex is not formed. This suggests that WDR34, as a subunit isolated from all other dynein subunits, is absolutely required for ciliogenesis. The authors address this clearly in their Discussion (subsection “Structural and functional asymmetry of the dynein-2 motor”). As this is in my opinion the most interesting point of the paper, the absence of mechanistic data to support this interesting finding leaves us with an unsatisfying open end.

*Reviewer #2:*

Vuolo et al. reported the function of two dynein-2 intermediate chains WDR34 and WDR60 in cilium formation and function. Using CRISPR-Cas9 to generate knockout cell lines, they uncovered that WDR34 but not WRD60 is essential for the extension of axonome and that both are involved in transition zone formation and bidirectional IFT. While the documentation of these phenotypes are interesting and valuable to the field, the deeper mechanistic insights may improve the current study.

1) Are the defects on transition zone the direct cause of WDR34 and WDR60 mutations? While the use of the individual fluorescence markers is informative, TEM is always the best solution to show the defects. Figure 1 is nice and convincing to show ciliary phenotype. Can authors point out the defects of transition zone from these TEM images? Will the loss of any other components in IFT-particle or IFT-motors generate the similar defects on transition zone formation?

2) How to interpret the distinct phenotypes between WDR34 and WDR60? Do they bind to different cargo molecules? The authors performed affinity purification and mass spectrometry analysis, did they find anything different between WDR34 and WDR60? In addition, how did they normalize the data for the quantification in Figure 9A.

3) Time-lapse fluorescence imaging will definitely improve the current study. Are WDR34 and WDR60 in the same moving particle or different or both? Will the loss of one affect the motility of the other? While the biochemistry is great, live imaging analysis and kymograph can be more direct and informative.

4) Can WDR34 and WDR60 be involved in other cellular processes that are mediated by dynein-1? Do the author notice any other defects?

5) For the ciliophathy-related mutations, the interpretation should be more cautious. For example, overexpression was used to introduce the point mutations into the genome. The dosage may compensate for the loss of activity. Ideally, knock-in should be used to introduce these mutations. Careful discussion is required for this caveat.

*Reviewer #3:*

This is an overall well-presented study using the RPE1 over expression system to look at the functions of the human WDR34 and WDR60 dyneins of Dynein-2. Data shows different functional effects of CRISPR mediated knockdown and recreation of human Jeune syndrome mutations suggesting a role for WDR34 in cilia extension and that WDR60 is essential for stability of Dynein-2. Some human mutations are shown to retain functionality. There is an intriguing effect of knockdown on the cilia transition zone. The study contains a lot of novel data but several areas need to be addressed to clarify some points.

1) Abstract stating 'Jeune syndrome is caused by defects in TZ architecture etc.' is not really accurate, since the study shows some of these disease mutations retain function but the transition zone markers were looked at only in KO cells. This should be removed or rephrased along the lines of the final Discussion sentences, to less dramatic claim.

2) Protein nomenclature is difficult for Dynein-2 with different names for the same subunit in different species. For the human genes/proteins using LIC3 (DYNC2LI1) and DHC2 (DYNC2H1) is confusing. It seems best to offer both species names, for clarity.

3) Results subsection “WDR34 or WDR60 play different roles in cilia function”, first paragraph. Are the CRISPR treated 'control' cells good cells for deciding whether there are off target effects? This is not clear – there could be different off target effects in different cells: so it would be helpful to show alongside some completely untreated cells in Figure 1—figure supplement 2. The text says 'all figures show WT cells where indicated. Indistinguishable results were obtained etc.' what is the meaning of WT and indistinguishable in this context? Is whole genome sequencing possible to confirm a lack of off target effects.

4) Figure 1B. WDR60 doesn't affect ciliogenesis, but we can see that Arl13b has a different distribution, this needs to be discussed. A control is missing for WDR34 in Figure 1C.

5) Figure 1H, how often is the bulged tip seen in WDR60 KO cells?

6) Several figures describe the cilia base e.g. in Figure 2, but staining for a marker for the basal body is needed, such as gamma tubulin – to confirm there is not a defect here explaining the changes seen in staining of some of the markers in the paper at the cilia base. Does a regular BB marker behave like normal in the mutant cells?

7) Figure 2G does not show any Golgi marker so reference to the 'Golgi pool' is rather circumspect, what is the proof that changes to this pool is what this is showing?

8) Figure 4A/B. I cannot see that SSTR3 is accumulating at the cilia tip here. Reduced levels yes, tip accumulation, not showing up in the figure resolution that I have. It's similar for 5HT6, not very clear in these images.

9) Referencing is needed for the publications describing the human mutations in WDR34 and WDR60 recreated here.

10) Figure 7A, where is the equivalent for WDR34?

11) A simple blot of the expressed mutant proteins that recreate the patient mutations in WDR34 and WDR60 is required, especially for the Q631*. This is standard requirement, to prove these mutated proteins are actually being expressed at the usual levels and therefore useable in experiments like the coIPs shown.

12) Figure 7B needs to show the double-mutant transfected cells immunofluorescence results as well. Figure 7C needs also to show the WDR60 KO cells IFT140 staining, mentioned in the text. In this figure call 'NT' the same thing in the panels and the graphs for full clarity i.e. in (B, Bi, C) and in (D, Di, E). Figure 7F indicate with arrow the FL and truncated WDR60 bands.

13) Figure 7—figure supplement 1B, C, how do we know that both mutations have been successfully co-expressed together in these cells.

14) Reference needed in the last paragraph of the subsection “Expression of wild-type and patient mutants of WDR60 and WDR34 in KO cells” for statement that NudCD3 interacts with Dynein-2.

15) Is the work in Figure 8 done in conjunction with serum starvation to stimulate cilia in the cells?

16) The experiment in Figure 9—figure supplement 1 is not crystal clear, edit the legend to clarify what was done.

17) The mass spectroscopy does not all fit the model shown in Figure 9. Explain why DYNLRB1 (in contrast to the text) and DYNLT3 do not fit the same trend of being reduced in WDR60 KO. The final Results sentence states a 'slight' reduction in DHC2 in WDR34-KO cells, this doesn't look correct-the change seems just as big as those claimed for some of the WDR60 KO cell reduced protein levels.

Levels of WDR60 in WDR34 KO cells and WDR34 in WDR60 KO cells look to be the same as WT approximately, whereas the model implies reduced WDR34 in WDR60 KO cells, not reflecting the data. Asante 2014 showed siRNA of WDR34 affects stability of WDR60 and vice versa – why is this not apparently reflected in this ms data?

18) The Discussion mentions that WDR34 may deliver to cilia a key factor in ciliogenesis – did the TMT mass spec data show any likely candidates for what this might be?

[Editors’ note: what now follows is the decision letter after the authors submitted for further consideration.]

Thank you for submitting your article "Dynein-2 intermediate 1 chains play crucial but distinct roles in primary cilia formation and function" for consideration by *eLife*. Your article has been reviewed by three peer reviewers, and the evaluation has been overseen by Andrew Carter as the Reviewing Editor and Anna Akhmanova as the Senior Editor. The reviewers have opted to remain anonymous.

The reviewers have discussed the reviews with one another and the Reviewing Editor has drafted this decision to help you prepare a revised submission.

Your manuscript reports an extensive characterisation of the phenotypes observed in ciliated cells when the dynein-2 intermediate chains (WDR60 and WDR34) are knocked out. The paper contains a number of important observations: most notably that dynein-2 function is required for the transition zone structure and that knocking out WDR60 and WDR34 have surprisingly different effects. The manuscript is a resubmission and the reviewers agree that you have significantly rearranged the results and text to make your points clearly.

The mechanistic insight has been improved by the inclusion of quantitative proteomics data. However the reviewers agreed that the manuscript would be stronger if some of your predictions could be directly tested. We suggest the live cell experiments in Essential revision 1 as a way of doing this. The other revisions (Essential revisions 2 – 4) can be addressed by changes to the text.

Essential revisions:

1) The authors should try to assess IFT particle velocity and trajectory on kymographs for the WDR34 and WDR60 KO cells e.g. by over-expression of IFT88-GFP. We appreciate that the WDR34 KO cells have few cilia and that the authors have already stated that live-cell imaging has been attempted but appeared to lack reproducibility. However, we feel that this additional data provides some insight into dynamic behaviour and supplements the steady-state data already presented in the manuscript. One obvious prediction would be that IFT-B proteins (IFT88, IFT57) would remain at the ciliary base/tip in any remaining cilia in WDR34 KO cells. However, the looser assembly of the dynein-2 motor and weaker interactions with IFTs in WDR60 KO cells would lead to increased IFT mobility.

2) While it is accepted in the author's rebuttal that TZ disruption maybe be a downstream effect rather than a primary consequence of WDR34/60 defects, this has not translated to the paper text, so this still needs to also be acknowledged in the Discussion section (subsection “Dynein-2 is required for transition zone composition”, last paragraph).

3) The WDR60 KO cells have intraciliary membraneous vesicles, but there is no direct evidence in the results that these are necessarily a consequence of a TZ defect, as strongly claimed here, that this is 'entirely consistent with a significant defect in the diffusion barrier'. This claim is made without any references to the literature for backup, so the Discussion section (subsection “Dynein-2 is required for transition zone composition”, first paragraph) needs correct referencing for this hypothesis, especially since the authors report that there are no TZ structural defects seen in the KO cells by EM. There are furthermore no reports of disrupted TZ in Jeune syndrome patients, as might be the implication from their including reference to patient mutations in this TZ Discussion section.

4) It still remains elusive how dynein-2 subunits contribute to transition zone formation. These KO cells lines were not genetically backcrossed as animals and TEM only limits to the certain number of cilia. The possibility that another mutation causes transition zone defects cannot be fully excluded. However, this is not the major breakthrough in this study, in comparison to subunit asymmetry. The author can discuss the caveats and tone down the role of dynein-2 subunits in transition zone.

---

## [Author Response]

[Editors’ note: the author responses to the first round of peer review follow.]Reviewer #1:Cilia assembly, maintenance, and functioning is dependent on bidirectional intraflagellar transport between its base and growing tip. Transport back to the base is carried out by cytoplasmic dynein-2 that unlike cytoplasmic dynein-1 has an asymmetric tail complex. The two different intermediate chains (ICs) WDR34 and WDR60 in this tail have been linked to cilia related disease. However a molecular characterization of the function of the asymmetric tail subunits remained absent.The authors show in human cells that depletion and patient mutations in both WDR60 and WDR34 lead to overlapping disturbances in ciliary membrane composition and transition zone integrity. Depletion of WDR60 disrupts the complete dynein complex, affecting retrograde transport, while cilia are still able to form. In contrast, WDR34 depletion does not disrupt the full dynein complex yet ciliogenesis is severely affected. With this the authors show that the different subunits have overlapping but also distinct roles that are not limited to retrograde transport.The findings in this manuscript are interesting and the data is solid. However, the absence of mechanistic data leaves this reviewer to contemplate whether it is suited for publication in eLife.

We do not consider that the summary above is not entirely accurate and apologize if this is down to our initial framing of our work. We *do* have molecular characterization from our proteomics data. We now extend our manuscript by including these data that show that there are not significant differences in binding to most components in the absence of WDR34 but instead, greater binding to some key components and a lack of localization to the basal body that suggests a stalled and mis-localized assembly of the axoneme extension machinery. We now include the proteomics data as a supplemental Excel file that illustrates this fully as well as including significant revision throughout the manuscript. We elaborate on this below.

1) The authors clearly show that the transition zone and membrane composition of the cilium are perturbed in relation to IC depletion. However, there is no mechanistic data for how the transition zone is perturbed. This leaves us with the question how far downstream the transition zone perturbation is from dynein perturbation. I would like to see how dynein depletion leads to disruption of the transition zone.

This would require acute interference with dynein-2 and while we have sought to achieve this using ciliobrevin and dynarrestin, we have not found these compounds effective in our cells without significant toxicity. RNAi as an example would not resolve this in terms of whether it is a direct dynein-2 function or an IFT function owing to the time frame of these experiments and evident IFT defects seen (Asante et al). We fully accept that TZ disruption might lie downstream of, for example, IFT defects We argue that what is significant here is that this perturbation could be a common underlying cause of ciliopathy phenotypes. Further analysis is beyond the scope of this study.

We have done the experiment asked and find no difference in the TZ in WDR34 or WDR60 depleted cells. These data are shown as Author response image 1 as we do not consider that they add to the manuscript. Our interpretation is that this is almost certainly owing to this being a depletion not knockout of these IC subunits.

**Author response image 1. respfig1:** (**A**) RNAi of WDR34 and WDR60 validated by immunoblotting with GAPODH as a loading control. (**B**) Immunofluorescence of TMEM67 and RPGRIP1L in WDR34 and WDR60 depleted cells. TMEM67 is seen at the base and within the cilium proximal to this in both control and depleted cells, RPGRIP1L is more tightly restricted to the base of the cilium in all cases. These are representative images, we have analysed multiple fields of view in each case using z-stacks and cannot identify any difference.

2) It is striking that, according to the MS data, the unassembled [WDR60 KO] dynein complex leads to a less severe phenotype than the partially intact [WDR34 KO] dynein complex. The partially assembled dynein that lacks WDR34 results in a ten-fold reduction in ciliated cells. In the MS data, HA-WDR34 pulls down all components less well in the [WDR60 KO] than the WT, showing the dynein complex is not formed.

This statement is not entirely accurate – the complex is formed but less well. Indeed, this less efficient association of the motor subunits is also reflected in a reduced association with several key IFT proteins. That said, this is clearly sufficient to enable axoneme extension unlike the case following deletion of WDR34.

This suggests that WDR34, as a subunit isolated from all other dynein subunits, is absolutely required for ciliogenesis. The authors address this clearly in their Discussion (subsection “Structural and functional asymmetry of the dynein-2 motor”). As this is in my opinion the most interesting point of the paper, the absence of mechanistic data to support this interesting finding leaves us with an unsatisfying open end.

This is the core of our argument which we now support with further data and enhanced explanation. Loss of WDR34 results in loss of DHC2 and LIC3 labelling at the centrosome. Our proteomics data show that these subunits remain capable of assembling a partial dynein-2 complex (missing WDR34) and that this can also bind to IFT and BBS proteins. Therefore, the absence of WDR34 results in mislocalization of dynein-2 and a failure of axoneme extension. In contrast, loss of WDR60 does not remove DHC2 from centrosomes and enables axoneme extension.

Reviewer #2:Vuolo et al. reported the function of two dynein-2 intermediate chains WDR34 and WDR60 in cilium formation and function. Using CRISPR-Cas9 to generate knockout cell lines, they uncovered that WDR34 but not WRD60 is essential for the extension of axonome and that both are involved in transition zone formation and bidirectional IFT. While the documentation of these phenotypes are interesting and valuable to the field, the deeper mechanistic insights may improve the current study.

While we understand the nature of this comment, we would not agree that we do not provide significant mechanistic insight within the manuscript, especially with the additional revisions.

1) Are the defects on transition zone the direct cause of WDR34 and WDR60 mutations? While the use of the individual fluorescence markers is informative, TEM is always the best solution to show the defects. Figure 1 is nice and convincing to show ciliary phenotype. Can authors point out the defects of transition zone from these TEM images?

There are no obvious defects from our TEM. That is why TEM alone is not sufficient, the defect is in transition zone composition (we use the phrase carefully) and accept that we should not have inferred defects in TZ structure. References to changes in TZ structure have been removed.

Will the loss of any other components in IFT-particle or IFT-motors generate the similar defects on transition zone formation?

To our knowledge a direct effect of any other IFT components on transition zone formation has not been described. Although often defects in IFT trafficking cause disruption of cilia structure and membrane protein trafficking that could be caused by a defective ciliary gating zone. Our work reveals for the first time a link between transition zone formation and dynein-2 motor. Future work will be focused on determining if this is a specific dynein-2 function or if it is dependent on some specific IFT components yet to be identified. To identify these components is beyond the scope of this work.

2) How to interpret the distinct phenotypes between WDR34 and WDR60? Do they bind to different cargo molecules? The authors performed affinity purification and mass spectrometry analysis, did they find anything different between WDR34 and WDR60? In addition, how did they normalize the data for the quantification in Figure 9A.

We now expand on our proteomics data as it is here that we do provide mechanistic insight. There are no obvious candidates missing from the interactomes. The differences like in the assembly of the dynein-2 complex and the strength of interaction with key partners, the IFT and BBS complexes. In 34KO, the dynein2 complex is formed but is non-functional. This could be that WDR34 is required for ATPase activity but exploring that is beyond our current means (although we have submitted a grant application to work on this in future). A simple explanation is that this complex is “stalled” in tight association with IFT and BBSome proteins. Our localization data show that this is not associated with the basal body. Therefore, axoneme extension requires a more dynamic interaction of these core complexes than is allowed in WDR34 KO cells.

In WDR60 KO cells, dynein-2 is less stably associated but can still function. Interactions with IFT proteins are also reduced. This is permissive for axoneme extension but not for fully functional IFT. This results in an accumulation of particles at the ciliary tip. Thus, a dynamic assembly of these key components is needed for axoneme extension.

Our next goal will be to develop our work to define the relationship between the relevant multiprotein complexes, dynein-2, kinesin-2, IFT-A, IFT-B and the BBSome. Analysis is highly complicated by the large size of these complexes, the labile nature of some of these complexes (notably IFT-A/B), and the difficulty in providing a robust and clearly defined outcome to such experiments. We would consider this beyond the scope of this current work.

3) Time-lapse fluorescence imaging will definitely improve the current study. Are WDR34 and WDR60 in the same moving particle or different or both? Will the loss of one affect the motility of the other? While the biochemistry is great, live imaging analysis and kymograph can be more direct and informative.

We thank the reviewer for the positive words on our biochemistry. Measuring IFT in RPE1 cells is not currently possible for us. We have tried extensively but cannot get the same quantitative data that is possible from other systems.

4) Can WDR34 and WDR60 be involved in other cellular processes that are mediated by dynein-1? Do the author notice any other defects?

We do not observe any defects in dynein-1-dependent processes. This is consistent with our previous work in this area (see (Palmer et al., 2009; Palmer et al., 2011)).

5) For the ciliophathy-related mutations, the interpretation should be more cautious. For example, overexpression was used to introduce the point mutations into the genome. The dosage may compensate for the loss of activity. Ideally, knock-in should be used to introduce these mutations. Careful discussion is required for this caveat.

We fully accept this caveat and discuss it (now more clearly) in the text. That said, our biochemical experiments demonstrate clearly a reduced association of WDR60 mutants with other dynein-2 subunits and with IFT140, IFT88, and IFT57.

Reviewer #3:This is an overall well-presented study using the RPE1 over expression system to look at the functions of the human WDR34 and WDR60 dyneins of Dynein-2. Data shows different functional effects of CRISPR mediated knockdown and recreation of human Jeune syndrome mutations suggesting a role for WDR34 in cilia extension and that WDR60 is essential for stability of Dynein-2. Some human mutations are shown to retain functionality. There is an intriguing effect of knockdown on the cilia transition zone. The study contains a lot of novel data but several areas need to be addressed to clarify some points.

We thank the reviewer for their positive framing of our work. We agree that the amount of novel data and defined outcomes that we present make our manuscript suitable for *eLife*. At the same time, we have tried to take care not to overinterpret data or over-extrapolate it in drawing our conclusions.

1) Abstract stating 'Jeune syndrome is caused by defects in TZ architecture etc.' is not really accurate, since the study shows some of these disease mutations retain function but the transition zone markers were looked at only in KO cells. This should be removed or rephrased along the lines of the final Discussion sentences, to less dramatic claim.

This sentence arises because some Jeune syndrome mutations result in complete loss of function of WDR34/WDR60. We do now state this more clearly in the Discussion and have rephrased the end of the Abstract.

2) Protein nomenclature is difficult for Dynein-2 with different names for the same subunit in different species. For the human genes/proteins using LIC3 (DYNC2LI1) and DHC2 (DYNC2H1) is confusing. It seems best to offer both species names, for clarity.

We now include both gene names in each instance but point out that these are not species names but come from an attempt to unify nomenclature ((Pfister et al., 2005) with which we were not involved).

3) Results subsection “WDR34 or WDR60 play different roles in cilia function”, first paragraph. Are the CRISPR treated 'control' cells good cells for deciding whether there are off target effects? This is not clear – there could be different off target effects in different cells: so it would be helpful to show alongside some completely untreated cells in Figure 1—figure supplement 2. The text says 'all figures show WT cells where indicated. Indistinguishable results were obtained etc.' what is the meaning of WT and indistinguishable in this context? Is whole genome sequencing possible to confirm a lack of off target effects.

We take the point here but argue that these are a better control than WT in most cases. WT cells are also shown in other figures and at no point have we detected any differences between WT and “CRISPR control” cells. We now state this clearly in the text (subsection “WDR34 or WDR60 play different roles in cilia function”).

4) Figure 1B. WDR60 doesn't affect ciliogenesis, but we can see that Arl13b has a different distribution, this needs to be discussed. A control is missing for WDR34 in Figure 1C.

We now discuss this in the subsection “WDR34 or WDR60 play different roles in cilia function”. WT cells and the CRISPR control cells are both controls for WDR34 KO. For the sake of space, we have not included a further control (no differences were seen).

5) Figure 1H, how often is the bulged tip seen in WDR60 KO cells?

This is common in both EM and in live cells (see Figure 4) and the text already cross-refers to this.

6) Several figures describe the cilia base e.g. in Figure 2, but staining for a marker for the basal body is needed, such as gamma tubulin – to confirm there is not a defect here explaining the changes seen in staining of some of the markers in the paper at the cilia base. Does a regular BB marker behave like normal in the mutant cells?

We have added co-staining of gamma tubulin with IFT88 and IFT54. No differences were observed in the localization of gamma-tubulin in KO cells compared with WT.

7) Figure 2G does not show any Golgi marker so reference to the 'Golgi pool' is rather circumspect, what is the proof that changes to this pool is what this is showing?

We have clarified this point in the text.

8) Figure 4A/B. I cannot see that SSTR3 is accumulating at the cilia tip here. Reduced levels yes, tip accumulation, not showing up in the figure resolution that I have. It's similar for 5HT6, not very clear in these images.

Perhaps we phrased this poorly and meant that the tip is enlarged and shows clear labeling for these markers. We have rephrased this in relation to Figure 4.

9) Referencing is needed for the publications describing the human mutations in WDR34 and WDR60 recreated here.

(McInerney-Leo et al., 2013) is the reference and is now included on in the first paragraph of the subsection “Expression of wild-type and patient mutants of WDR60 and WDR34 in KO cells.

10) Figure 7A, where is the equivalent for WDR34?

The WDR34 mutant data have been removed for clarity.

11) A simple blot of the expressed mutant proteins that recreate the patient mutations in WDR34 and WDR60 is required, especially for the Q631*. This is standard requirement, to prove these mutated proteins are actually being expressed at the usual levels and therefore useable in experiments like the coIPs shown.

This is shown in Figure 7G. Full lengths WDR60 and truncated Q631* are indicated with a red asterisk.

12) Figure 7B needs to show the double-mutant transfected cells immunofluorescence results as well. Figure 7C needs also to show the WDR60 KO cells IFT140 staining, mentioned in the text. In this figure call 'NT' the same thing in the panels and the graphs for full clarity i.e. in (B, Bi, C) and in (D, Di, E). Figure 7F indicate with arrow the FL and truncated WDR60 bands.

We have now modified Figure 7 extensively and the double transfected cells are included as Figure 7—figure supplement 1. We do not consider there to be sufficient space, or these data to have sufficient impact, to include these data within Figure 7.

13) Figure 7—figure supplement 1B, C, how do we know that both mutations have been successfully co-expressed together in these cells.

In Figure 7—figure supplement 1 we show these data. All cells express HA-WDR60-Q631* and these were then transfected with HA-WDR60-T749M (Figure 7—figure supplement 1B). Thus, the latter indicates the presence of both (Figure 7—figure supplement 1C).

14) Reference needed in the last paragraph of the subsection “Expression of wild-type and patient mutants of WDR60 and WDR34 in KO cells” for statement that NudCD3 interacts with Dynein-2.

Now included (Asante et al.).

15) Is the work in Figure 8 done in conjunction with serum starvation to stimulate cilia in the cells?

Included in the subsection “The stability of dynein-2 complex in WDR34 and WDR60 KO cells”.

16) The experiment in Figure 9—figure supplement 1 is not crystal clear, edit the legend to clarify what was done.

The legend has now been clarified.

17) The mass spectroscopy does not all fit the model shown in Figure 9. Explain why DYNLRB1 (in contrast to the text) and DYNLT3 do not fit the same trend of being reduced in WDR60 KO. The final Results sentence states a 'slight' reduction in DHC2 in WDR34-KO cells, this doesn't look correct-the change seems just as big as those claimed for some of the WDR60 KO cell reduced protein levels.Levels of WDR60 in WDR34 KO cells and WDR34 in WDR60 KO cells look to be the same as WT approximately, whereas the model implies reduced WDR34 in WDR60 KO cells, not reflecting the data.

We have reworked this section extensively including the presentation of the data.

Asante 2014 showed siRNA of WDR34 affects stability of WDR60 and vice versa – why is this not apparently reflected in this ms data?

WDR34 KO does affects the stability of WDR60 and vice versa. This is shown in western blot Figure 8A. The proteomics experiments analyse the stability of the full complex using overexpressed HA-WDR60 and HAWDR34 as reporters.

18) The Discussion mentions that WDR34 may deliver to cilia a key factor in ciliogenesis – did the TMT mass spec data show any likely candidates for what this might be?

Essentially no and this reframes our arguments about mechanistic insight here.

[Editors' note: the author responses to the re-review follow.]

Essential revisions:1) The authors should try to assess IFT particle velocity and trajectory on kymographs for the WDR34 and WDR60 KO cells e.g. by over-expression of IFT88-GFP. We appreciate that the WDR34 KO cells have few cilia and that the authors have already stated that live-cell imaging has been attempted but appeared to lack reproducibility. However, we feel that this additional data provides some insight into dynamic behaviour and supplements the steady-state data already presented in the manuscript. One obvious prediction would be that IFT-B proteins (IFT88, IFT57) would remain at the ciliary base/tip in any remaining cilia in WDR34 KO cells. However, the looser assembly of the dynein-2 motor and weaker interactions with IFTs in WDR60 KO cells would lead to increased IFT mobility.

As you note, we commented in our previous correspondence that our attempts to assay IFT were not reproducible. We have been trying to improve this, but we simply have not been able to. Arl13b looked like a good marker but we now have good evidence that the apparent IFT observed in kymographs was in fact largely an artefact since the measured velocity changed as a function of the acquisition rate (frames per second) We believe that the kymograph analysis is falsely connecting traces due to the high particle density, leading to spurious results. We have also tried to use IFT88 tagged with a variety of fluorescent proteins (mEmerald and EYFP) again to no avail. Either the cytosolic background is too high or the overexpression itself inhibits ciliogenesis. In RPE1 cells the cilia are submerged within the cell. This complicates any analysis in very low expressing cells owing to the cytosolic background and lack of amenability of our cells to TIRF microscopy. While this method can work in some cell types, we have not been able to apply this to our RPE1 (from ATCC). Kazu Nakayama's lab has shown some nice IFT of EGFP-DYNC2LI1 in RPE1 cells (and using TIRF; Figure 7 of their paper (Hamada et al., 2018)) however, they do not show any quantification of these data, possibly due to difficulties in obtaining sufficient numbers of cells. Notably, this probe does not enter cilia, at least at detectable levels, in WDR60 KO cells. This is shown with our own labelling of endogenous protein, Figure 8B of our manuscript.

We have also included the following statement in the Results section:

“We have tried to measure IFT in these cells using a variety of markers including Arl13b- and IFT88-fusions but have not been able to derive reliable quantitative data.”

Therefore, we have not been able to address this query. In the interests of responsible research and reproducibility, we are not willing to include our Arl13b data or any other data we have in their present form. We would also add that we do not necessarily think that the “looser assembly of the dynein-2 motor and weaker interactions with IFTs in WDR60 KO cells would lead to increased IFT mobility”. We show tip accumulation in WDR60 KO cells and as such one might expect less, not more IFT mobility. Thus, we don’t consider that these data will be definitive.

2) While it is accepted in the author's rebuttal that TZ disruption maybe be a downstream effect rather than a primary consequence of WDR34/60 defects, this has not translated to the paper text, so this still needs to also be acknowledged in the Discussion section (subsection “Dynein-2 is required for transition zone composition”, last paragraph).

We apologise that his did not translate to the text with enough clarity. Our data are consistent with the concept that the role of dynein-2 might be in either assembling or maintaining the transition zone. This could, and indeed is quite likely to be, through its known role in IFT. We have now addressed this in the relevant section of the text with the following amendment:

“Our data do not discriminate between direct or indirect roles for dynein-2 in either building or maintaining the ciliary transition zone. […] Recent data from the Blacque lab using IFT-A mutants (Scheidel and Blacque, 2018) and from the Leroux labs using a temperature sensitive allele of the dynein-2 heavy chain (Jensen et al., personal communication), both using *C. elegans*, provides support for this interpretation.”

3) The WDR60 KO cells have intraciliary membraneous vesicles, but there is no direct evidence in the results that these are necessarily a consequence of a TZ defect, as strongly claimed here, that this is 'entirely consistent with a significant defect in the diffusion barrier'. This claim is made without any references to the literature for backup, so the Discussion section (subsection “Dynein-2 is required for transition zone composition”, first paragraph) needs correct referencing for this hypothesis, especially since the authors report that there are no TZ structural defects seen in the KO cells by EM. There are furthermore no reports of disrupted TZ in Jeune syndrome patients, as might be the implication from their including reference to patient mutations in this TZ Discussion section.

As with our comments above, we in fact favour a model in which the TZ defects seen are a result of impaired IFT. We have amended the text in this section (subsection “Loss of dynein-2 intermediate chains results in perturbed transition zone composition”) to better reflect the concerns raised here and a more considered evaluation of our data, including making the point that is well made here that there have been no structural defects described in cells derived from Jeune syndrome patients. Indeed, we have removed the sentence containing he phrase “entirely consistent with a significant defect in the diffusion barrier” entirely.

4) It still remains elusive how dynein-2 subunits contribute to transition zone formation. These KO cells lines were not genetically backcrossed as animals and TEM only limits to the certain number of cilia. The possibility that another mutation causes transition zone defects cannot be fully excluded. However, this is not the major breakthrough in this study, in comparison to subunit asymmetry. The author can discuss the caveats and tone down the role of dynein-2 subunits in transition zone.

We now include this caveat in the Discussion. Notably we have also changed the heading of this section in the Discussion to “Loss of dynein-2 intermediate chains results in perturbed transition zone composition”.